**Carbon geochemistry of plankton-dominated samples in the Laptev and East Siberian**
**shelves: contrasts in suspended particle composition**
Tesi Tommaso [1,2,3], Marc C. Geibel [1,2], Christof Pearce [2,4,5], Elena Panova [6], Jorien E. Vonk [7],
Emma Karlsson[1,2], Joan A. Salvado[1,2], Martin Kruså[1,2], Lisa Bröder [1,2], Christoph Humborg
[1,2], Igor Semiletov[6,8,9], Örjan Gustafsson [1,2]
[1] Department of Environmental Science and Analytical Chemistry (ACES), Stockholm
University
[2] Bolin Centre for Climate Research, Stockholm University
[3] Institute of Marine Sciences, National Research Council (ISMAR-CNR)
[4] Department of Geological Sciences, Stockholm University, Sweden
[5] Department of Geoscience, Aarhus University, Denmark
[6] Tomsk Polytechnic University
[7] Vrije Universiteit Amsterdam (VU)
[8] Pacific Oceanological Institute FEB RAS
[9] University of Alaska Fairbanks

**Abstract**

Recent Arctic studies suggest that sea-ice decline and permafrost thawing will affect phytoplankton dynamics and stimulate heterotrophic communities. However, in what way the plankton composition will change as the warming proceeds remains elusive. Here we investigate the chemical signature of the plankton-dominated fraction of particulate organic matter (POM, >10μm) collected along the Siberian shelf. POM (>10μm) samples were analysed using molecular biomarkers (CuO oxidation and $IP_{25}$) and dual-carbon isotopes ($\delta^{13}C$ and $\Delta^{14}C$). In addition, surface water chemical properties were integrated with the POM (>10μm) dataset to understand the link between plankton composition and environmental conditions.

$\delta^{13}C$ and $\Delta^{14}C$ exhibited a large variability in the POM (>10μm) distribution while the content of terrestrial biomarkers was negligible~~while terrestrial biomarkers showed a negligible input from terrestrial sources~~. In the Laptev Sea (LS) ~~open waters~~, $\delta^{13}C$ and $\Delta^{14}C$ fingerprint ~~of POM (>10μm)~~ suggested a heterotrophic environment in which ~~that~~ dissolved organic carbon (DOC) from the Lena river was the primary source of metabolizable carbon. Within the Lena plume, terrestrial DOC likely became part of the food web via bacteria on which other heterotrophic communities (e.g. dinoflagellates) fed on. ~~indicating, thus, a heterotrophic environment.~~ Moving eastwards toward the sea-ice dominated East Siberian Sea (ESS), the system became progressively more autotrophic. Comparison between $\delta^{13}C$ of POM (>10μm) samples and $CO_2aq$ concentrations revealed that the carbon isotope fractionation increased moving toward the easternmost and most productive stations.

In a warming scenario characterized by enhanced terrestrial DOC release (thawing permafrost) and progressive sea-ice decline, heterotrophic conditions might persist in the LS while the nutrient-rich Pacific inflow will likely stimulate greater ~~ESS~~ primary productivity in

the ESS. The contrasting trophic conditions will result in a sharp gradient in $\delta^{13}C$ between the LS and ESS similar to what documented in our semi-synoptic study.

## 1. Introduction

The progressive reduction of sea-ice extent in the Arctic Ocean is indisputable evidence of modern global warming (Comiso et al., 2008; Ding et al., 2017; Kwok and Rothrock, 2009). The unprecedented decline of sea-ice is expected to alter several aspects of the Arctic marine ecology such as plankton abundance and its temporal distribution (Arrigo et al., 2008). For instance, recent studies suggest that the increase of solar irradiance will stimulate greater primary productivity in summer while the prolonged ice-free conditions will develop a second algal bloom in early fall, which is a distinctive feature of only lower latitudes (Ardyna et al., 2014; Lalande et al., 2009; Lalande et al., 2014). The phytoplankton communities are expected to profoundly change towards a higher contribution from open water phytoplankton at the expense of sea-ice assemblages (Fujiwara et al., 2014). Taken together, a greater productivity in the ice-free or marginal ice zone compare to the multi-year ice system, is also expected to lead to greater carbon uptake and settling export of organic carbon from the surface to deeper strata of the Arctic Ocean (Gustafsson and Andersson, 2012).

Sea-ice decline will also affect the water-air gas exchange, currents and river plume dispersion which, in turn, exert large control on the surface water chemical/physical properties (Aagaard and Carmack, 1989; Ardyna et al., 2014; Lalande et al., 2014). On top of this, destabilization of permafrost soils and the terrestrial cryosphere will result in enhanced particulate and dissolved carbon input to the Arctic Ocean (Frey and Smith, 2005; Vonk et al., 2012). As a result, the geochemical signature of both autotrophic and heterotrophic plankton communities is also expected to change as the warming proceeds. However, how the warming

will ultimately affect the marine geochemical signal is poorly understood. This study seeks a better understanding of the chemical composition of plankton that dominates regions of the Arctic Ocean characterized by different sea-ice coverages, nutrient availability and riverine influence. In particular, we focus on the carbon isotope fingerprint (i.e. $\delta^{13}C$ and $\Delta^{14}C$) of plankton that grows in ice-covered and ice-free Marginal Ice Zone (MIZ) regimes on the Siberian margin. The motivation behind investigating the chemical fingerprint of plankton from different regimes is to provide a better understanding of the carbon signature for direct applications to carbon studies of both modern systems and paleo-reconstructions. In particular, the isotope composition of marine OC finds several applications in climate, ecology and carbon source apportionment studies. For example, stable carbon isotopes of marine phytoplankton are used for paleo-$p$CO$_2$ reconstructions over geological time scales (Hoins et al., 2015; Pagani et al., 1999; Popp et al., 1999; Rau, 1994). The $\delta^{13}C$ signature also provides a solid tool for marine food web and ecosystem structure investigations (Dunton et al., 2006; Iken et al., 2005; Kohlbach et al., 2016). Furthermore, dual-carbon isotope mixing models ($\delta^{13}C$ and $\Delta^{14}C$) are commonly used to quantify the relative proportion of marine and various allochtonous sources (e.g., permafrost soil) in both contemporary and paleo-reconstructed carbon cycling of the Arctic (Karlsson et al., 2016; Tesi et al., 2016; Vonk et al., 2012; Vonk et al., 2014).

With this overarching goal in mind, here we investigate the >10 μm fraction of particulate organic matter (POM) in ice-covered and ice-free MIZ regimes of the Siberian Arctic Shelf during the SWERUS-C3 expedition (July-August 2014) (Fig. 1). The plankton-dominated POM samples collected throughout the ca. 4,500 km long cruise track were characterized using bulk parameters (OC, $\delta^{13}C$ and $\Delta^{14}C$) and biomarkers (highly branched isoprenoids, IP$_{25}$; CuO oxidation products). In addition, continuous measurements of dissolved $CO_2$ ($CO_{2aq}$) and its stable carbon isotope composition ($\delta^{13}C_{CO2}$) were performed

during the campaign (Humborg et al., 2017) and used for a direct comparison with the
chemical composition of the POM fraction.

**2. Study region**
The Laptev Sea and the East Siberian Sea are shallow epicontinental seas in the
Russian Arctic separated by the New Siberian Islands (Fig. 1). Sea-ice cover lasts for most
part of the year over the shelf. Late spring/summer is characterized by the seasonal sea-ice
retreat coupled with river freshet which supplies large amount of terrestrial carbon in the form
of particulate and dissolved matters (Karlsson et al., 2016; Salvadó et al., 2016; Sánchez-
García et al., 2011). The Lena (523 km3/y), Indigirka (54 km3/y), and Kolyma (48 km3/y) are
the major rivers (Gordeev, 2006). During the ice-free season, the Lena plume can be traced in
the outer-shelf of the Laptev Sea (Fichot et al., 2013; Salvadó et al., 2016; Sánchez-García et
al., 2011) while Pacific inflow from the Bering strait affects further east the East Siberia
margin (Semiletov et al., 2005).  The Pacific inflow exerts control on the nutrient balance as it
supplies ~~nitrates and nitrites~~phosphorous and silicate to an otherwise nutrient-depleted region
(Anderson et al., 2011; Semiletov et al., 2005). Another important source of particulate
material to the continental margin is the Pleistocene Ice Complex Deposit (ICD) entering the
ocean via coastal erosion (Lantuit et al., 2011; Vonk et al., 2012) which is the dominant
carbon source between the Kolyma river and the Lena river (Vonk et al., 2012).

**3. Methods**
**3.1 POM (<10 μm) sampling**
Seawater was pumped from a stainless steel inlet on the hull of the icebreaker *Oden*
positioned at 8 m below the sea surface. The inlet system is tested and further described in
Sobek and Gustafsson (2004) and Gustafsson et al. (2005). Figure 1a and 1b show the regions
covered to harvest each POM (>10 μm) sample with their location shown as time-averaged
position. The particulate material was retained via a large volume filtration apparatus using a
10-μm Nitex® (nylon) mesh placed in a 29.3 cm filter holder. After collection, filtered
particulate material was transferred in pre-clean HDPE tubes by rinsing the Nitex® filters
with MilliQ water. Samples were kept frozen throughout the expedition. In the lab, samples
were transferred in pre-cleaned Falcon® tubes (rinsed with 0.1M HCl) and gently centrifuged
to remove the supernatant. The residual particulate material was frozen and subsequently
freeze-dried prior to biogeochemical analyses.

3.2 Bulk carbon isotopes and biomarker analyses

Organic carbon (OC) and stable carbon isotope ($\delta^{13}C$) analyses were carried out on

acidified samples (Ag capsules, HCl, 1.5M) to remove the carbonate fraction (Nieuwenhuize
et al., 1994). Analyses were performed using a Thermo Electron mass spectrometer directly
coupled to a Carlo Erba NC2500 Elemental Analyzer via a Conflo III (Department of
Geological Sciences, Stockholm University). OC values are reported as weight percent
(%d.w.) whereas stable isotope data are reported in the conventional $\delta^{13}C$ notation (‰). The
analytical error for $\delta^{13}C$ was lower than ±0.1‰ based on replicates. Acidified (HCl, 1.5 M)
samples for radiocarbon abundance were analysed at the US-NSF National Ocean Science
Accelerator Mass Spectrometry (NOSAMS) facility (Woods Hole Oceanographic Institution,
Woods Hole, USA). Radiocarbon data are reported in the standard $\Delta^{14}C$ notation (‰).

Alkaline CuO oxidations were carried out using an UltraWAVE Milestone microwave

as described in Tesi et al. (2014). Briefly, about 2 mg of OC was oxidised using CuO under
alkaline (2N NaOH) and oxygen-free conditions at 150 °C for 90 min in teflon tubes. After
the oxidation, known amounts of recovery standards (trans-cinnamic acid and ethylvanillin)
were added to the solution. The NaOH solutions were then acidified to pH 1 with
concentrated HCl and extracted with ethyl acetate. Extracts were dried and redissolved in
pyridine. CuO oxidation products were quantified by GC-MS in full scan mode (50-650 m/z).
Before GC analyses, the CuO oxidation products were derivatized with bis(trimethylsilyl)
trifluoroacetamide+1% trimethylchlorosilane at 60°C for 30 min. The compounds were
separated chromatographically in a 30m×250 µm DB5ms (0.25 µm thick film) capillary GC
column, using an initial temperature of 100°C, a temperature ramp of 4°C/min and a final
temperature of 300°C. Lignin phenols (terrestrial biomarkers) were quantified using the
response factors of commercially available standards (Sigma-Aldrich) whereas the rest of the
CuO oxidation products were quantified by comparing the response factor of trans-cinnamic
acid. Lignin-derived reaction products include vanillyl phenols (V=vanillin, acetovanillone,
vanillic acid), syringyl phenols (S=syringealdehyde, acetosyringone, syringic acid) and
cinnamyl phenols (C=p-coumaric acid, ferulic acid). In addition to lignin, cutin-derived
products (hydroxyl fatty acids) were used to trace the land-derived input (Goñi and Hedges,
1990; Tesi et al., 2010). Other CuO oxidation products include para-hydroxybenzene
monomers (P-series), benzoic acids (B-series) and short-chain fatty acids (FA-series) which
can have both terrestrial and marine origin (Goñi and Hedges, 1995; Tesi et al., 2010).

The sea-ice proxy $IP_{25}$ (mono-unsaturated highly branched isoprenoid (HBI) alkene)

was quantified according to Belt et al. (2012). $IP_{25}$ producers are a minor (<5%) fraction of
the total sea-ice taxa which are, however, ubiquitous in pan-Arctic sea-ice. Species include
*Pleurosigma stuxbergii var. rhomboide*, *Haslea crucigeroides* (and/or *Haslea spicula*) and
*Haslea kjellmanii* (Brown et al., 2014a). Briefly, lipids were extracted via sonication using a
dichloromethane/methanol solution (2:1 v/v × 3). Prior to the extraction, two internal
standards (7-hexylnonadecane, 7-HND and 9- octylheptadecene, 9-OHD) were added to
permit quantification of $IP_{25}$ (monounsaturated highly branched isoprenoid) following
analysis via GC-MS. Total lipid extracts (TLEs) were dried under $N_2$ after removing the water
excess with anhydrous NaSO$_4$. Dry TLEs were redissolved in dichloromethane and the non-
polar hydrocarbon fraction was purified using open column chromatography (deactivated
SiO$_2$) and hexane as eluent. Saturated and unsaturated n-alkanes were further separated using
10% AgNO$_3$ coated silica gel using hexane and dichloromethane, respectively.
Quantification of IP$_{25}$ was carried out in SIM mode (*m/z* 350.3) as described in Belt et
al. (2012). The GC was fitted with a 30m×250 µm DB5ms (0.25 µm thick film) capillary GC
column. Initial GC oven temperature was set to 60°C followed by a 10°C/min ramp until a
final temperature of 310°C (hold time 10 min).

**3.3. Microscope images of plankton**
High resolution digital images were taken with an Environmental Scanning Electron
Microscope (ESEM) Philips XL30 FEG in high voltage (15kV) and magnification 250X.
Samples were further studied for identification of diatoms and dinoflagellates using a
transmitted light microscope (Leitz Laborlux 12 Pol) equipped with differential interference
contrast optics at 1000X magnification. Microscope slides were prepared using settling
chambers to achieve an even distribution of particles on the cover glass, regardless of size and
shape ~~Warnock and Sherer (2014).~~(Warnock and Scherer, 2015).

**3.4 Sea-ice data**
Daily AMSR2 sea-ice extent and concentration maps were provided by the Institute of
Environmental Physics, University of Bremen, Germany (Spreen et al., 2008) as GeoTIFF
files (ftp://seaice.uni-bremen.de).

**3.5 Statistics**
We used two-tailed T-test (homoscedasticity) and Welch T-test (heteroskedasticity) to
assess whether the differences between open waters and sea-ice dominated waters were
statistically significant. For this study, significance level (alpha) was set at 0.01.

**4. Surface water conditions during the SWERUS-C3 expedition**

Before discussing the chemical composition of the POM (>10 μm), here we briefly
introduce the different environmental conditions encountered throughout the cruise track. The
surface water data presented in this section were pulled together from previous studies which
provide an in-depth analysis of the surface water properties during the SWERUS-C3
expedition in 2014 (Humborg et al., 2017; Salvadó et al., 2016) (Table 2). For this study,
continuous $p$CO$_2$aq and $\delta^{13}$C$_{CO2}$ data (Humborg et al., 2017) were averaged to match the
water sampling stations allowing for a direct comparison with DOC and salinity data (Fig. 2)
(Supplementary Material).
Summer 2014 was consistent with the long-term downward trend in Arctic sea-ice
extent. The strongest anomalies were observed in the LS which experienced the most
northerly sea-ice shift since satellite observations began in 1979 (National Snow and Sea Ice
Data Center, NSIDC. http://nsidc.org/data). Unpublished data). In general, sea-ice displayed a
strong gradient over the study region going from ice-free conditions in the outer LS to ice-
dominated waters in the outer ESS (Fig.1.) Three snapshots of the sea-ice extent and
concentrations (.i.e. at the beginning, in the middle and at the end of the sampling) is shown in
Fig.1. Furthermore, Table 1 reports the averaged sea-ice concentrations encountered during
the collection of each sample.
The surface water salinity exhibited a longitudinal trend characterized by low values
in the outer LS while the sea-ice dominated ESS waters showed relatively higher values (Fig.
2a; Table 2). However, the highest salinity values were measured in the westernmost stations
resulting in a sharp gradient in the LS. The low surface water salinities in the outer LS are
most likely the result of both Lena river input and sea-ice thawing (Humborg et al., 2017) that
started in late May (Janout et al., 2016).
The highest DOC concentrations were measured in the mid-outer LS in the surface
water plume affected by Lena River runoff (Fig.2b; Table 2). Overall, DOC concentrations
followed the plume dispersion with high DOC concentrations corresponding to low salinities
(Fig. 2). Carbon stable isotopes ($\delta^{13}$C) and terrestrial biomarkers (of the solid-phase extracted
DOC fraction; Salvado et al., 2016) further confirmed the influence of terrestrial DOC in the
outer LS, while the land-derived input progressively decreased moving eastward.
$p$CO$_2$aq concentrations exhibited a typical estuarine pattern over the study region
(Humborg et al., 2017) (Fig. 2d; Table 2). Low salinity waters in the outer LS showed above
atmospheric CO$_2$ concentrations (i.e., supersatur${\text{d}}$oversaturation) while surface waters
below sea-ice exhibited undersaturated concentrations. The most depleted $\delta^{13}$C$_{CO2}$ values
were measured off the Lena river mouth (Fig. 2e; Table 2). Being relatively rich in land-
derived material, it is likely that respired terrestrial OC within the Lena river plume exerted
control on the CO$_2$ isotopic signature and concentration (Humborg et al., 2017).
Finally, nutrient distribution revealed nitrate (NO$_3$) and nitrite (NO$_2$) depletion in
surface waters throughout the cruise track (Humborg et al., 2017) in comparison with the
Arctic Ocean gateways such as the Bering strait. Here, nutrient concentrations in surface
waters are two-order of magnitude higher compared the our study region (Torres-Valdés et
al., 2013). Phosphate (PO$_4$) exhibited rather low concentrations in the outer LS and relatively
higher concentrations below the sea-ice in the outer ESS (Humborg et al., 2017) likely
reflecting the inflow of nutrient-rich Pacific waters (Anderson et al., 2011; Semiletov et al.,
2005; Torres-Valdés et al., 2013).

**Formatted:** Font: Italic

## 5. Results and discussion

### 5.1 Source of the POM (>10 μm) fraction

The Arctic Ocean off northern Siberia receives large quantities of dissolved and particulate terrestrial organic carbon via continental runoff and coastal erosion (Alling et al., 2010; Dittmar and Kattner, 2003; McClelland et al., 2016; Sánchez-García et al., 2011; Semiletov et al., 2013; Vonk et al., 2012). The land-derived material that does not settle in the coastal zone further travels across the continental margin reaching out to the outer-shelf region resuspended within the benthic nepheloid layer or in suspension within the surface river plume (Fichot et al., 2013; Sánchez-García et al., 2011; Wegner et al., 2003). Another fraction of terrestrial material can travel across the Siberian margin trapped in fast ice (Dethleff, 2005). Considering the potential allochthonous contribution, we addressed to what extent terrestrial organic material affects the POM (>10μm) fraction by quantifying the concentration of lignin phenols and C16-18 hydroxy fatty acids (cutin-derived products). These biomarkers are exclusively formed by terrestrial vegetation and, thus, serve as tracers of land-derived material in the marine environment (Amon et al., 2012; Bröder et al., 2016b; Feng et al., 2015).

Upon CuO alkaline oxidation the POM (>10μm) samples yielded only traces of lignin phenols while the cutin-derived products were not detected (Fig. 3). Other oxidation products in high abundance included saturated and mono-unsaturated short chain fatty acids (C12-18FA), para-hydroxy phenols, benzoic acids and dicarboxylic acids. These other reaction products are ubiquitous in both marine and terrestrial environments but they are predominant in plankton-derived material, especially short-chain fatty acids (Goñi and Hedges, 1995). When compared with active-layer permafrost soils and ice-complex deposits (Tesi et al., 2014), POM (>10μm) samples displayed a distinct CuO fingerprint dominated by short chain fatty acids (Fig. 3), consistent with the typical CuO products yields by phytoplankton batch

cultures upon CuO alkaline oxidation (Goñi and Hedges, 1995). SEM images further
corroborated the abundance of marine plankton detritus in the POM (>10μm) fraction while
lithogenic particles (clastic material) appeared to be sporadic in all samples.
The OC content (% d.w.) of the POM (>10um) fraction decreased eastwards showing
high concentrations in the LS and relatively low values in the ESS (Table 1; $p<0.01$ T-test).
However, in terms of absolute concentration in the water column (μC/l), the highest levels
were generally observed in the sea-ice covered region (Table 1; Fig. 4a; $p<0.01$ T-test).
Qualitative analyses by SEM and transmitted-light microscopy highlight important
differences in plankton assemblages which reflect different timing of the plankton blooms
which can explain these differences in concentration. Specifically, the open-water LS stations
exhibited a low degree of plankton diversity and were largely dominated by a bloom of
heterotrophic dinoflagellate cysts (*Protoperidinium* spp) (Fig. 5a; Table 3). Moving towards
the ice-dominated regions, diatoms become the prevailing species. Dominant diatom genera
include *Chaetoceros spp.* (dominant diatom in several stations), *Thalassiosira spp.*,
*Rhizosolenia spp.*, *Coscinodiscus spp.*, *Asteromphalus spp.*, *Navicula spp.* as well as sea-ice
species such as *Fragilariopsis cylindrus* and *Fragilariopsis oceanica* (Fig. 5b,c; Table 3).
Moored optical sensors deployed in the LS shelf recorded the sea-ice retreat in 2014
and found no sign of pelagic under-ice blooms despite available nutrients while high
chlorophyll concentrations were detected immediately after the ice retreated in late May
(Janout et al., 2016). The ice-edge blooms lasted for about 2 weeks according to the high
resolution chlorophyll time-series (Janout et al., 2016). Thus, our post-bloom sampling in the
LS essentially captured an oligotrophic environment dominated by heterotrophic
dinoflagellate cysts (i.e, Protoperidinium spp) which likely fed on phytodetritus and river-
derived organic material. Such conditions are fairly consistent with the relatively low carbon
contents (μgC/L) observed in LS waters (Fig. 4a).

The Arctic sea-ice biomarker IP25 (Fig. 4b) further highlights the different regimes observed in ice-free and ice-dominated surface waters. IP25 is a proxy of sea-ice based on a highly branched mono-unsaturated isoprenoid alkene found in some sea-ice diatoms which, however, generally account for 5% of the total sea-ice taxa (Belt et al., 2007; Brown et al., 2014b). The IP25 concentrations varied by several orders of magnitude over the study area showing low concentrations in the open-water western region while the sea-ice dominated surface waters to the east exhibited high concentrations especially at station 31b (Fig. 4b; Table 1) ; $p<0.01$ Welch T-test). The fact that IP25 was still detectable throughout the ice-free outer LS suggests that the proxy captured the signal of the sea-ice retreat that occurred shortly before the sampling at the end of May/early June (Janout et al., 2016). Alternatively, the IP25 could have been advected from nearby sea-ice dominated regions.

**5.2. Dual carbon isotopes: $\delta^{13}$C and $\Delta^{14}$C**

$\delta^{13}$C and $\Delta^{14}$C of the POM (>10μm) samples exhibited a distinctive longitudinal trend across the study area between LS and ESS (Fig. 4c,d) ($p<0.01$ T-test) . Depleted $\delta^{13}$C values characterized the LS open waters ranging from -28.1 to -24.7‰ (Fig. 4c). Although within the range of terrestrially-derived material, our CuO oxidation data (i.e. trace of lignin phenols and absence of cutin-derived products) suggest that the "light" isotopic composition in the LS might instead reflect the plankton assemblage dominated by heterotrophic dinoflagellate cysts as previously described (e.g., *Protoperidinium* spp; Fig. 5a). More specifically, heterotrophic dinoflagellates can adapt their metabolism depending on the substrate available (e.g., diatoms and bacteria). Several studies have shown that terrestrial DOC greatly promotes bacteria biomass production which in turn stimulates the growth of heterotrophic dinoflagellates (Carlsson et al., 1995; Purina et al., 2004; Wikner and Andersson, 2012). Thus, in these conditions, allochthonous terrestrial DOC is actively recycled by bacteria and transferred to

dinoflagellates which explains, thus, the depleted $\delta^{13}$C values observed in the river-dominated
samples (Carlsson et al., 1995).
The modern radiocarbon fingerprint of the Lena DOC discharge is consistent with
$\Delta^{14}$C signature of the POM (>10μm) fraction in the LS (up to +99 ‰), supporting the
importance of terrestrial DOC as a carbon source for the food web in the river plume (Fig. 4d
and 6). By contrast, comparison with other potential carbon sources which include the Lena
river particulate organic carbon, surface sediments, Pleistocene coastal Ice-Complex Deposit
and Pacific DIC inflow reveals a different (more depleted) radiocarbon fingerprint (Fig. 6). It
is also import to highlight that the DOC within the Lena plume is one/two-order of magnitude
higher than the particulate carbon pool supporting, thus, our hypothesis (Humborg et al.,
2017; Salvadó et al., 2016).
Moving towards the ice-dominated ESS, surface waters progressively becaome more
autotrophic and productive (Humborg et al., 2017) while the POM (>10μm) exhibited a wide
$\delta^{13}$C signature ranging from -28.6 to -21.2‰ (Fig. 4c). The most depleted values were
observed across the transition zone between open-waters and sea-ice. Visual inspections of
these samples revealed large abundance of the centric diatom *Chaetoceros* spp. (spores and
vegetative cells; St22, Fig. 5b) while lignin and cutin data indicated, a negligible input of
land-derived material. Primary factors determining the fractionation of stable carbon isotopes
in phytoplankton are several and include $CO_2$aq concentration, $\delta^{13}$Caq, growth rate, cell size,
cell shape, light and nutrient availability (Gervais and Riebesell, 2001; Laws et al., 1997b;
Popp et al., 1998; Rau et al., 1996). Our understanding about isotopic fractionation has been
historically achieved via laboratory experiments designed to test each factor under controlled
conditions. In natural environments, however, different factors can compete with each other,
sometimes in opposite directions. Yet, the existing knowledge about surface water properties
during the expedition (Humborg et al., 2017) can provide important constraints for the
isotopic signal interpretation.
For example, comparison with continuous $\delta^{13}C$-$CO_2$aq and $p$$CO_2$aq data measured
throughout the cruise track - time-averaged to match the large volume filtration along the
cruise track (Table 1) - suggested a negligible role exerted by $\delta^{13}C$-$CO_2$aq (Fig. 7b) while
$p$$CO_2$aq concentration correlated with the $\delta^{13}C$ of the POM (>10µm) fraction ($r^2$=0.72;
$p$<0.01) (Fig. 7a). Such a relationship fits with the general model according to which a low
demand (i.e., low growth rate) and high supply (i.e., abundant $CO_2$aq) favour high
fractionation and vice versa (Laws et al., 1997a; Laws et al., 1995; Wolf-Gladrow et al.,
1999).

During the expedition, surface water properties (i.e. $O_2$ and $CO_2$, Table 2) (Humborg
et al., 2017) suggest that the productivity in the outer ESS increases moving eastward, as
commonly observed, likely due to the Pacific inflow (Anderson et al., 2011; Semiletov et al.,
2005). As a result, the wide range of plankton $\delta^{13}C$ over the ESS can be explained in terms of
two different regimes: (a) in the transition zone between open waters and sea-ice, the
productivity was low but $p$$CO_2$aq was ~~supersaturated oversaturated~~ while (b) in the
easternmost ESS, productivity was high but $p$$CO_2$aq was depleted (Fig. 7b). The former
regime favours fractionation while the latter does not (Fig. 7b). Different diatom assemblages
can also be another factor to consider although the phytoplankton diversity observed over ESS
can be considered rather small (e.g. *Chaetoceros spp.* dominant in most of the samples)
compared to the wide range of $\delta^{13}C$ observed (i.e., from -28.8 to -21.6) (Table 3).
The POM (>10µm) fraction in the sea-ice dominated ESS exhibited slightly - but
consistently - depleted $\Delta^{14}C$ values ranging from -62 to -49 ‰ (Fig. 4d). This region is
affected by the inflow of Pacific waters whose DIC exhibits, however, a modern $\Delta^{14}C$
signature (Griffith et al., 2012) (Fig. 6). By contrast, these results suggest the influence from
an aged carbon pool. As the ESS remains covered by sea-ice for most of the year, it is
possible that the sea-ice hampers the gas exchange with the atmosphere and acts as a lid by
trapping $CO_2$ which derives from the breakdown of sedimentary organic material (Anderson
et al., 2009; Semiletov et al., 2016), which might have such ages (Bröder et al., 2016a; Vonk
et al., 2012). In these conditions, the pre-aged $CO_2$ accumulates underneath the sea-ice and is
subsequently incorporated during carbon fixation by the phytoplankton. While
~~supersaturated~~oversaturated bottom waters were extensively documented in the region with
important consequences on the local DIC (Anderson et al., 2009; Pipko et al., 2009), more
work is clearly needed to understand if early diagenesis in sediments can also affect the
radiocarbon signature of the $CO_2$aq underneath the sea-ice. Alternatively, the slightly depleted
radiocarbon signature might indicate the presence of pre-aged terrestrial organic carbon (Fig.
6) in the POM (>10μm) samples, not reflected in the lignin and cutin tracers (Fig. 3).
However, it would then remain elusive why such an aged land-derived influence was not
visible in the river-dominated LS waters while it affected the sea-ice dominated region.

Taken together, our results indicate that the dual-carbon isotope fingerprint is highly

affected by the trophic conditions (heterotrophic *vs* autotrophic) as well as the extent of
primary productivity. In a warming scenario characterized by sea-ice retreat (Arrigo et al.,
2008; Comiso et al., 2008) and enhanced terrestrial input from land as result of hydrology and
permafrost destabilization (Frey and Smith, 2005; Vonk et al., 2012), the geochemical
composition of plankton will likely change as the warming proceeds.

**6. Conclusions**

Analyses of large-volume filtrations of plankton-dominated >10 μm particle samples

revealed a high degree of heterogeneity in the dual carbon isotope signature ($\delta^{13}C$ and $\Delta^{14}C$)
between ice-free waters (Laptev Sea) and the ice-covered region (East Siberian Sea).
Our results suggest a heterotrophic environment in the outer LS open waters where the
$\delta^{13}C$ depleted river DOC is transferred to relatively higher trophic levels via microbial
incorporation in the river plume. Moving eastwards towards the ice-dominated outer ESS,
surface waters became progressively more autotrophic. Here, the isotopic fractionation
appears to follow the phytoplankton growth *vs* $CO_2$ demand model according to which carbon
fractionation decreases at high growth and low $CO_2$ concentrations. As a result, the transition
between open-waters and sea-ice exhibited more depleted $\delta^{13}C$ values compared to the
productive easternmost stations. Radiocarbon signatures were slightly depleted over the whole
sea-ice dominated area. This raises the question whether the sea-ice hampers the gas exchange
with the atmosphere and trap the $CO_2$ sourced from reactive sedimentary carbon pools.
In a warming scenario, it is likely that the oligotrophic ice-free LS will be dominated
by heterotrophic metabolism fuelled by terrestrially-derived organic material (i.e., Lena
input). In these conditions, the dual-carbon isotope signature of the heterotrophic plankton
will essentially reflect the terrestrial fingerprint. In the ESS, which receives the inflow of the
nutrient-rich Pacific waters, ice-free conditions will enhance light penetration. This in turn
might further stimulate phytoplankton growth with important implications in terms of $CO_2$
depletion and resulting low isotope fractionation. Altogether, this will result in a sharp
compositional gradient (e.g. $\delta^{13}C$) between LS and ESS similar to what captured in our semi-
synoptic study.

**Acknowledgements**
We thank the *I/B Oden* crew and the Swedish Polar Research Secretariat staff. This
study was supported by the Knut and Alice Wallenberg Foundation (KAW contract
2011.0027), the Swedish Research Council (VR contract 621-2007-4631 and 621-2013-
5297), European Research Council (ERC-AdG CC-TOP project #695331 to Ö.G.). T. Tesi
additionally acknowledges EU financial support as Marie Curie fellow (contract no. PIEF-
GA-2011-300259). J.A. Salvadó acknowledges EU financial support as a Marie Curie grant
(contract no. FP7-PEOPLE-2012-IEF; project 328049). I. Semiletov acknowledges financial
support from the Russian Government (grant No. 14, Z50.31.0012/03.19.2014) and the
Russian Foundation for Basic Research (nos. 13-05-12028 and 13-05-12041), and E. Panova
from the Russian Scientific Foundation (grant no. 15-17-20032). We thank the Arctic Great
Rivers Observatory (NSF-1107774) for providing DOC and POC river data
(www.arcticgreatrivers.org). This is ISMAR publication ID n.1940.

















**Table 1. Chemical composition of the POM (>10μm) fraction and continuous CO₂aq measurements***

| ID | Time averaged latitude (N) | Time averaged longitude (E) | Mean sea-ice percentage (%) | POM (>10μm) concentration (mg/l) | OC (d.w.) | $\delta^{13}C$ (‰) | $\Delta^{14}C$ (‰) | IP25 (ng/gOC) | averege $CO_2$aq (ppm)* | average $\delta^{13}C$-$CO_2$aq (‰)* |
|---|---|---|---|---|---|---|---|---|---|---|
| ST4 | 81.68 | 105.96 | 98.4 | 6 | 18.2 | -26.7 | n.d. | n.d. | 323 | -10.9 |
| ST5 | 80.47 | 114.07 | 98.7 | 15 | 42.6 | -27.6 | n.d. | n.d. | 322 | -11.0 |
| ST6 | 78.86 | 125.22 | 82.2 | 1 | 51.7 | -26.6 | 99 | n.d. | 325 | -10.8 |
| ST7 | 77.88 | 126.62 | 0.0 | 11 | 43.1 | -25.7 | n.d. | 88 | 350 | -10.7 |
| ST8 | 77.16 | 127.32 | 0.0 | 17 | 30.9 | -26.7 | 41 | n.d. | 391 | -10.5 |
| ST9 | 76.78 | 125.83 | 0.0 | 3 | 31.5 | -27.9 | 30 | 48 | 385 | -10.5 |
| ST10 | 76.90 | 127.81 | 0.0 | 11 | 40.9 | -24.7 | n.d. | n.d. | 349 | -11.0 |
| ST11 | 77.12 | 126.66 | 0.0 | 13 | 29.6 | -28.1 | 27 | 13 | 428 | -10.7 |
| ST22 | 77.67 | 144.63 | 0.0 | 20 | 11.3 | -28.8 | n.d. | 95 | 394 | -11.0 |
| ST23 | 76.43 | 147.53 | 0.0 | 6 | 7.6 | -28.5 | -50 | n.d. | 394 | -11.2 |
| ST24 | 76.42 | 149.84 | 34.4 | 19 | 11.9 | -26.8 | -62 | 368 | 374 | -11.1 |
| ST25 | 76.62 | 152.03 | 96.7 | 23 | 19.5 | -25.7 | -31 | 465 | 263 | -10.8 |
| ST26 | 76.14 | 157.85 | 96.2 | 109 | 30.8 | -24.2 | -30 | 217 | 316 | -10.9 |
| ST27 | 75.00 | 161.03 | 91.5 | 41 | 23.3 | -23.0 | n.d. | 256 | 299 | -11.1 |
| ST28 | 74.63 | 161.98 | 86.3 | 28 | 15.5 | -23.8 | n.d. | n.d. | 214 | -11.3 |
| ST29 | 73.61 | 169.72 | 79.3 | 31 | 14.7 | -23.2 | -50 | 518 | 184 | -11.3 |
| ST30 | 75.61 | 174.01 | 66.7 | 43 | 22.6 | -27.0 | n.d. | n.d. | 304 | -10.5 |
| ST31A | 75.85 | 174.41 | 75.6 | 30 | 10.9 | -21.6 | -62 | 1911 | 182 | -10.6 |
| ST31B | 74.26 | 173.74 | 63.5 | 15 | 4.6 | -23.3 | n.d. | 783 | n.d. | n.d. |
| ST32 | 73.56 | 176.06 | 51.8 | 21 | 11.3 | -24.5 | -58 | 131 | n.d. | n.d. |
| ST33 | 72.35 | -175.14 | 0.0 | 20 | 15.5 | -23.5 | n.d. | 473 | n.d. | n.d. |
| ST34 | 73.28 | -173.05 | 28.7 | 76 | 13.4 | -21.6 | -52 | 970 | n.d. | n.d. |
| ST35 | 75.21 | -172.05 | 53.9 | 24 | 14.3 | -24.2 | n.d. | 268 | n.d. | n.d. |

n.d = not determined
*Humborg et al. (2017)











**Table 2. Surface water (0-20 m) chemical and physical properties during the SWERUS-C3 expedition***

| | Salinity | Temperature | DIC | DOC | POC | δ$^{13}$C-DIC | NO$_2$-NO$_3$ | PO$_4$ |
|---|---|---|---|---|---|---|---|---|
| | | ºC | µmol kg$^{-1}$ | µmol kg$^{-1}$ | µmol kg$^{-1}$ | ‰ | µmol kg$^{-1}$ | µmol kg$^{-1}$ |
| | median | median | median | median | median | median | median | median |
| Outer LS shelf (0-20 m) | 32.87 | 3.84 | 2139 | 149.1 | 7.9 | 0.75 | 0.21 | 0.27 |
| LS shelf break (0-20 m) | 33.56 | 0.57 | 2114 | 91.5 | 10.1 | 1.10 | 0.26 | 0.15 |
| Outer ESS shelf (0-20 m) | 29.45 | -1.33 | 1969 | 84.2 | 10.7 | 1.14 | 0.25 | 0.97 |
| ESS shelf break (0-20 m) | 28.23 | -1.32 | 1979 | 73.7 | 4.6 | 1.47 | 0.11 | 0.59 |
| | mean | mean | mean | mean | mean | mean | mean | mean |
| Outer LS shelf (0-20 m) | 31.17 | 3.40 | 2119 | 179.8 | 7.9 | 0.58 | 0.60 | 0.29 |
| LS shelf break (0-20 m) | 33.42 | 0.96 | 2111 | 97.5 | 10.0 | 1.10 | 0.61 | 0.16 |
| Outer ESS shelf (0-20 m) | 28.95 | -0.05 | 1949 | 95.8 | 11.9 | 1.26 | 0.26 | 0.95 |
| ESS shelf break (0-20 m) | 28.27 | -1.31 | 1975 | 72.0 | 4.6 | 1.49 | 0.12 | 0.60 |
| | s.d. | s.d. | s.d. | s.d. | s.d. | s.d. | s.d. | s.d. |
| Outer LS shelf (0-20 m) | 3.22 | 2.38 | 89 | 66.3 | 1.7 | 0.50 | 0.91 | 0.11 |
| LS shelf break (0-20 m) | 0.70 | 2.07 | 23 | 21.2 | 1.7 | 0.11 | 0.74 | 0.06 |
| Outer ESS shelf (0-20 m) | 1.41 | 2.28 | 75 | 30.2 | 4.6 | 0.49 | 0.12 | 0.19 |
| ESS shelf break (0-20 m) | 0.53 | 0.04 | 49 | 3.2 | 0.3 | 0.08 | 0.03 | 0.02 |

*data from Humborg et al. (2017) and Salvadó et al. (2016)















**Table 3. Qualitative plankton characterization of selected POM (>10μm) samples**

| ID | Region | Diatoms | Dinoflagellates | Other species |
|---|---|---|---|---|
| ST6 | LS | Few *Coscinodiscus* | None observed | |
| ST9 | LS | None observed | Few *Protoperidinium* | |
| ST11 | LS | None observed | Abundant *Protoperidinium* | |
| ST22 | LS-ESS | Abundant *Chaetoceros*, few *Rhizosolenia*, *Thalassiosira* | None observed | |
| ST25 | LS-ESS | High diversity. Abundant *Chaetoceros, few Rhizosolenia, Coscinodiscus, Thallasiosira, Asteromphalus, Navicula* | None observed | Silicoflagellate |
| ST31A | ESS | High diversity. Abundant *Chaetoceros, few Rhizosolenia, Thallasiosira, Bacterosira, Navicula* | None observed | |
| ST31B | ESS | High diversity. Few *Chaetoceros, Thallasiosira, Fragilariopsis* | Few *Protoperidinium* | |
| ST34 | ESS | Abundant *Chaetoceros*, few *Thalassiosira, Navicula* | Few *Protoperidinium* | |


















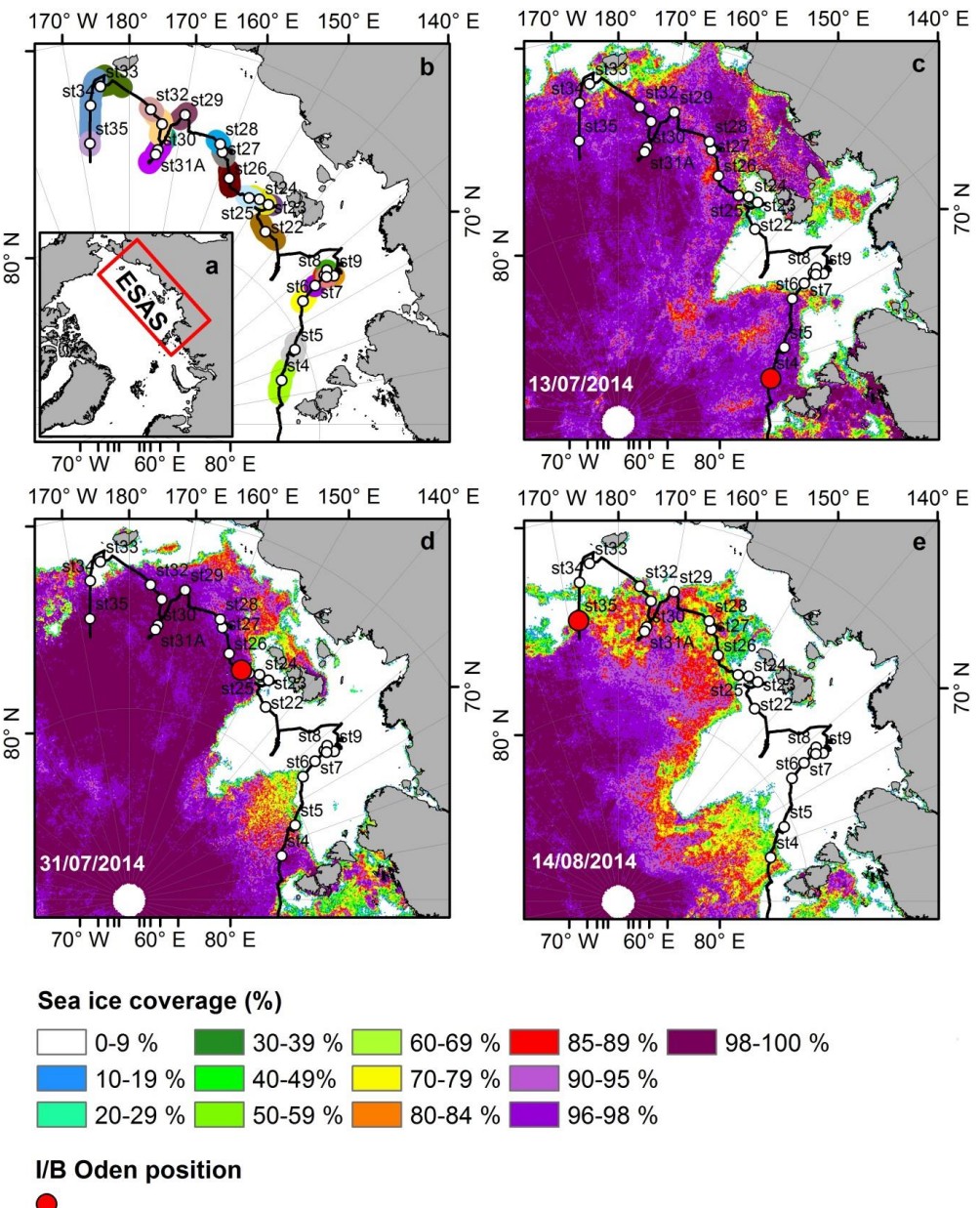


**Fig. 1** (a) The study area in the East Siberian Arctic Shelf. (b) Time-averaged position during the large-volume filtration (circles) of the POM (>10μm) samples. Shaded coloured areas show the sampling area covered to harvest each POM (>10μm) sample. Sea-ice extent and concentration at the beginning (c), in the middle (d) and at the end (e) of the sampling campaign. The ship position is shown by a filled red circle.

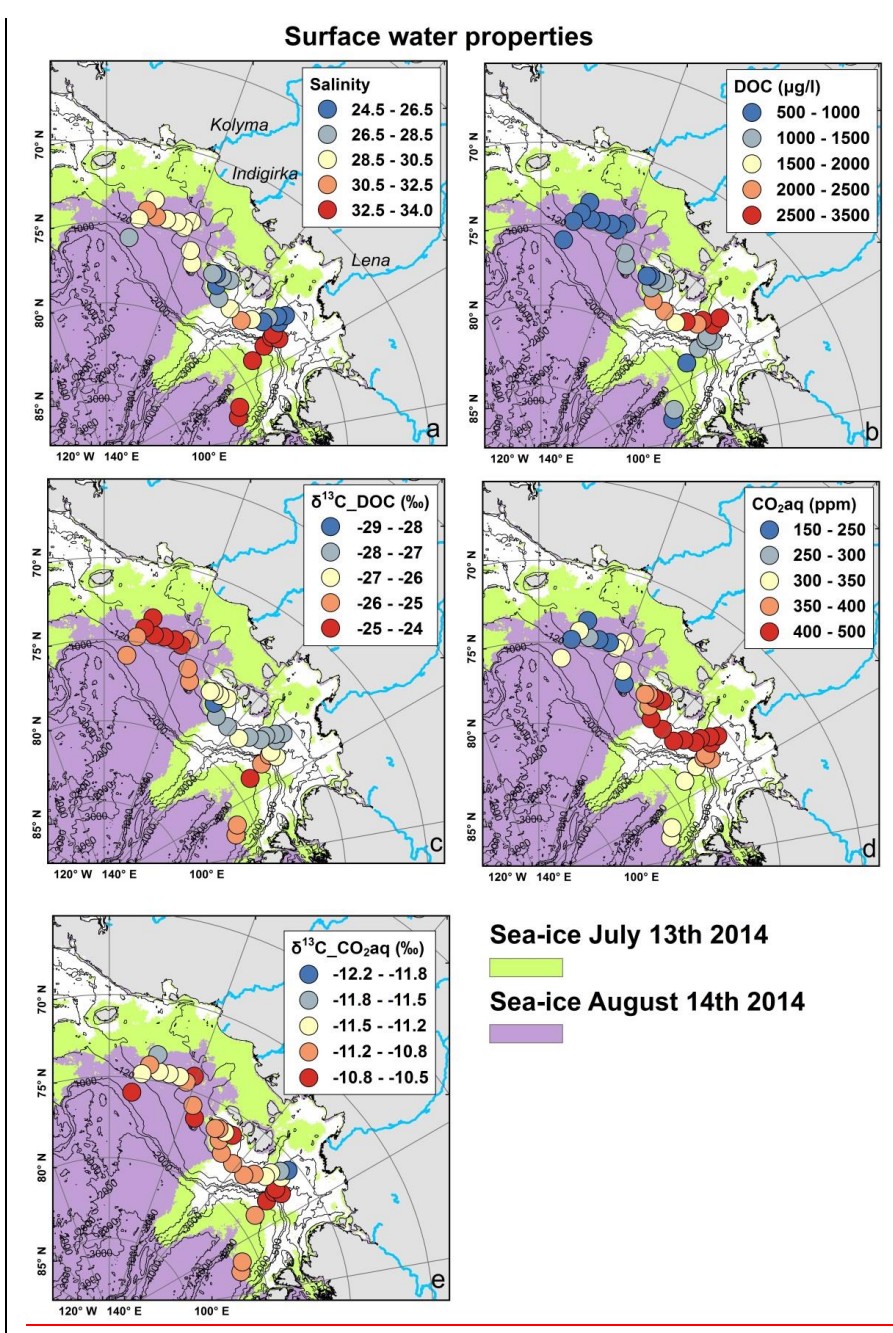

**Fig.2** Surface water properties. (a) Salinity. (b) DOC. (c) $\delta^{13}$C-DOC. (d) $CO_2$aq. (e) $\delta^{13}$C-$CO_2$aq. Shaded areas show the sea-ice extent at the beginning (13/07/2014) and at the end of the sampling campaign (14/08/2014) (Humborg et al., 2017; Salvadó et al., 2016).

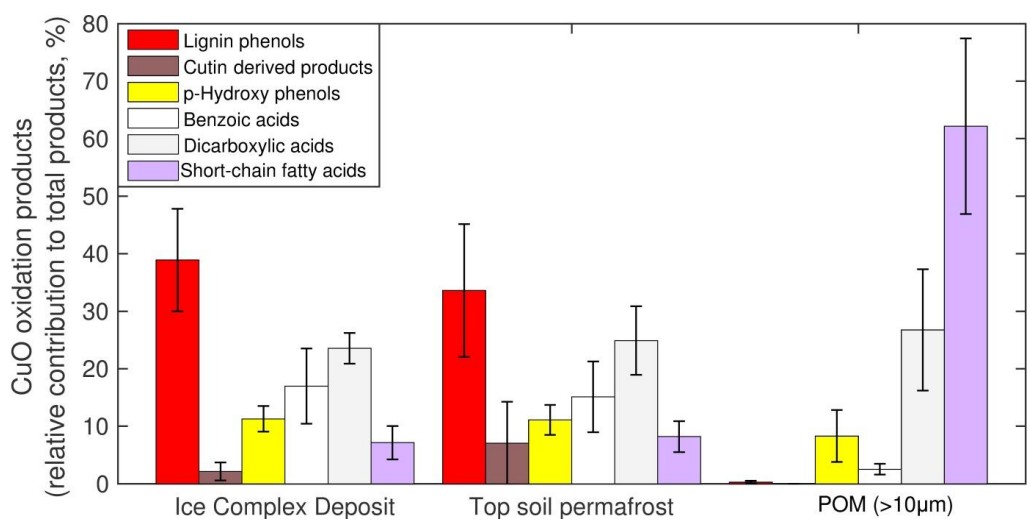





**Fig.3** Alkaline CuO fingerprint of top-soil permafrost samples (Tesi et al., 2014), Pleistocene
Ice Complex Deposit (Tesi et al., 2014) and POM (>10μm) fraction (this study). The plot
displays the relative proportion products yield upon alkaline CuO oxidation. The error bar
refers to the natural variability of each dataset










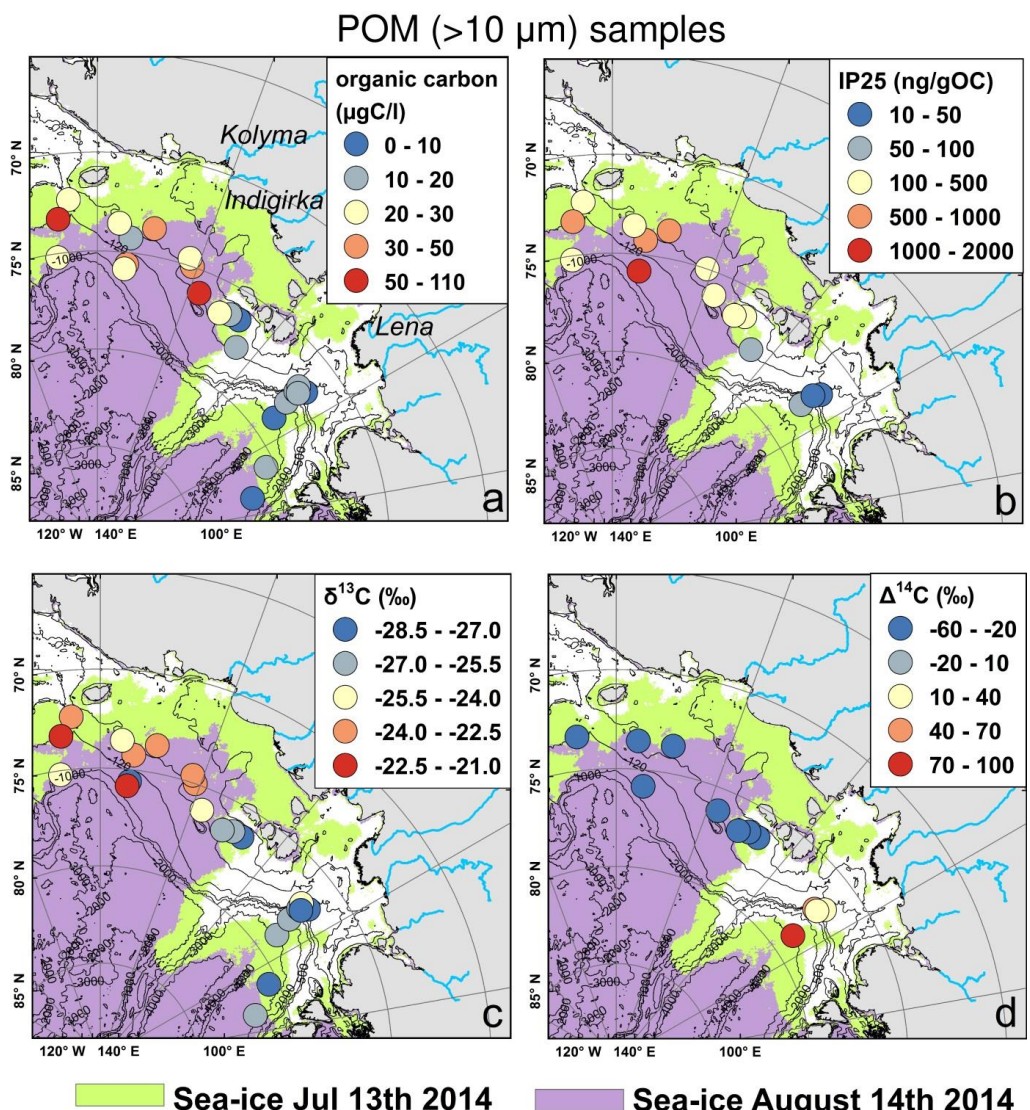

**Sea-ice Jul 13th 2014**    **Sea-ice August 14th 2014**

**Fig. 4** POM (>10μm) composition (a) Organic carbon concentration. (b) IP25 (mono-
unsaturated highly branched isoprenoid. (c) $\delta^{13}$C. (d) $\Delta^{14}$C. Shaded areas show the sea-ice
extent at the beginning (13/07/2014) and at the end of the sampling campaign (14/08/2014).



ST11 - Laptev Sea

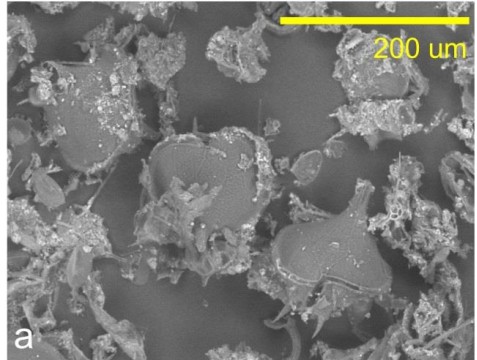

ST22 - Laptev Sea / East Siberian Sea

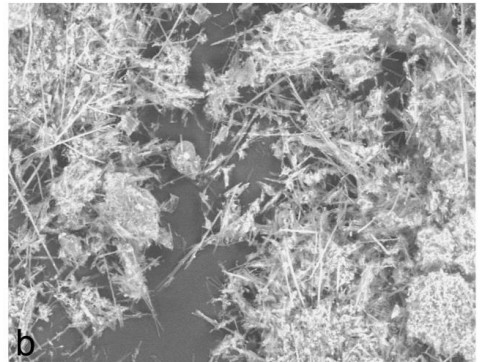

ST34 - East Siberian Sea

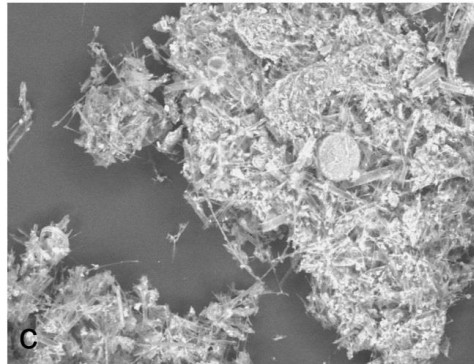



**Fig. 5** SEM images. (a) ST-11: Dinoflagellates (*Protoperidinium* spp.) in open-waters of the
Laptev Sea. (b) ST22: Diatoms, mostly spines (setae) of *Chaetoceros* spp. in the transition
between Laptev Sea and East Siberian Sea. (c) ST-34: Diatoms from sea-ice dominated
waters in the East Siberian Sea

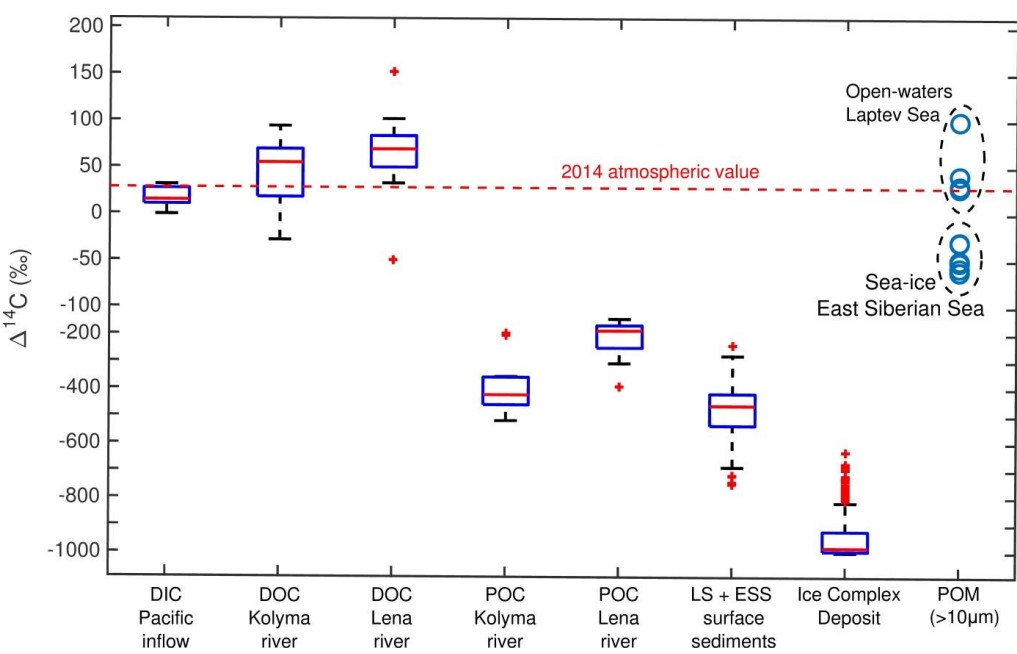




**Fig. 6** Radiocarbon signature of inorganic and organic carbon pools. Whisker plots of

radiocarbon values for different inorganic and organic carbon sources from the literature,

compared to the outer Laptev Sea and outer East Siberian Sea (blue circles, this study). Solid

lines show the median, the box limits display the 25[th] and 75[th] percentiles while the crosses

show the outliers. Source: DIC (Griffith et al., 2012), DOC-Kolyma (2009-2014), DOC-Lena

(2009-2014), POC-Kolyma (2009-2011), POC-Lena (2009-2011)

(www.arcticgreatrivers.org), Laptev Sea and Eastern Siberia Sea surface sediments (Salvadó

et al., 2016; Vonk et al., 2012) and Ice Complex Deposit (Vonk et al., 2012).






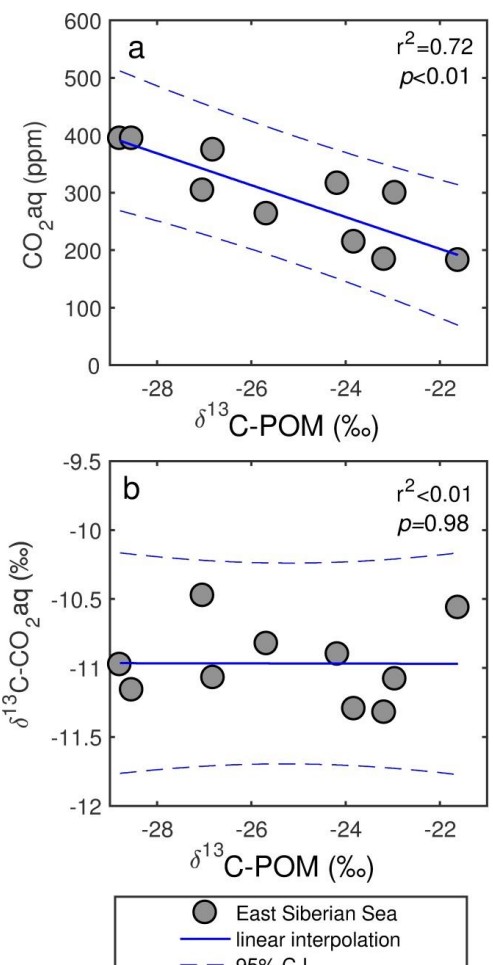

**Fig. 7** Correlations (a) $CO_2aq$ vs $\delta^{13}C$ (POM (>10μm) fraction) and (b) $\delta^{13}C$-$CO_2aq$ vs $\delta^{13}C$ in
the East Siberian Sea (filled circles). The solid line shows the linear interpolation while the
dashed line shows the 95% confidence intervals.

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

**Abstract**

Recent Arctic studies suggest that sea-ice decline and permafrost thawing will affect

phytoplankton dynamics and stimulate heterotrophic communities. However, in what way the
plankton composition will change as the warming proceeds remains elusive. Here we
investigate the chemical signature of the plankton-dominated fraction of particulate organic
matter (POM, >10μm) collected along the Siberian shelf. POM (>10μm) samples were
analysed using molecular biomarkers (CuO oxidation and $IP_{25}$) and dual-carbon isotopes
($\delta^{13}C$ and $\Delta^{14}C$). In addition, surface water chemical properties were integrated with the POM
(>10μm) dataset to understand the link between plankton composition and environmental
conditions.

$\delta^{13}C$ and $\Delta^{14}C$ exhibited a large variability in the POM (>10μm) while the content of

terrestrial biomarkers was negligible. In the Laptev Sea (LS), $\delta^{13}C$ and $\Delta^{14}C$ fingerprint
suggested a heterotrophic environment in which dissolved organic carbon (DOC) from the
Lena river was the primary source of metabolizable carbon. Within the Lena plume, terrestrial
DOC likely became part of the food web via bacteria on which other heterotrophic
communities (e.g. dinoflagellates) fed on.  Moving eastwards toward the sea-ice dominated
East Siberian Sea (ESS), the system became progressively more autotrophic. Comparison
between $\delta^{13}C$ of POM (>10μm) samples and $CO_2aq$ concentrations revealed that the carbon
isotope fractionation increased moving toward the easternmost and most productive stations.

In a warming scenario characterized by enhanced terrestrial DOC release (thawing

permafrost) and progressive sea-ice decline, heterotrophic conditions might persist in the LS
while the nutrient-rich Pacific inflow will likely stimulate greater primary productivity in the
ESS. The contrasting trophic conditions will result in a sharp gradient in $\delta^{13}C$ between the LS
and ESS similar to what documented in our semi-synoptic study.

**1. Introduction**

The progressive reduction of sea-ice extent in the Arctic Ocean is indisputable evidence of modern global warming (Comiso et al., 2008; Ding et al., 2017; Kwok and Rothrock, 2009). The unprecedented decline of sea-ice is expected to alter several aspects of the Arctic marine ecology such as plankton abundance and its temporal distribution (Arrigo et al., 2008). For instance, recent studies suggest that the increase of solar irradiance will stimulate greater primary productivity in summer while the prolonged ice-free conditions will develop a second algal bloom in early fall, which is a distinctive feature of only lower latitudes (Ardyna et al., 2014; Lalande et al., 2009; Lalande et al., 2014). The phytoplankton communities are expected to profoundly change towards a higher contribution from open water phytoplankton at the expense of sea-ice assemblages (Fujiwara et al., 2014). Taken together, a greater productivity in the ice-free or marginal ice zone compare to the multi-year ice system, is also expected to lead to greater carbon uptake and settling export of organic carbon from the surface to deeper strata of the Arctic Ocean (Gustafsson and Andersson, 2012).

Sea-ice decline will also affect the water-air gas exchange, currents and river plume dispersion which, in turn, exert large control on the surface water chemical/physical properties (Aagaard and Carmack, 1989; Ardyna et al., 2014; Lalande et al., 2014). On top of this, destabilization of permafrost soils and the terrestrial cryosphere will result in enhanced particulate and dissolved carbon input to the Arctic Ocean (Frey and Smith, 2005; Vonk et al., 2012). As a result, the geochemical signature of both autotrophic and heterotrophic plankton communities is also expected to change as the warming proceeds. However, how the warming will ultimately affect the marine geochemical signal is poorly understood. This study seeks a better understanding of the chemical composition of plankton that dominates regions of the Arctic Ocean characterized by different sea-ice coverages, nutrient availability and riverine

influence. In particular, we focus on the carbon isotope fingerprint (i.e. $\delta^{13}C$ and $\Delta^{14}C$) of
plankton that grows in ice-covered and ice-free Marginal Ice Zone (MIZ) regimes on the
Siberian margin. The motivation behind investigating the chemical fingerprint of plankton
from different regimes is to provide a better understanding of the carbon signature for direct
applications to carbon studies of both modern systems and paleo-reconstructions. In
particular, the isotope composition of marine OC finds several applications in climate,
ecology and carbon source apportionment studies. For example, stable carbon isotopes of
marine phytoplankton are used for paleo-$p$CO$_2$ reconstructions over geological time scales
(Hoins et al., 2015; Pagani et al., 1999; Popp et al., 1999; Rau, 1994). The $\delta^{13}C$ signature also
provides a solid tool for marine food web and ecosystem structure investigations (Dunton et
al., 2006; Iken et al., 2005; Kohlbach et al., 2016). Furthermore, dual-carbon isotope mixing
models ($\delta^{13}C$ and $\Delta^{14}C$) are commonly used to quantify the relative proportion of marine and
various allochtonous sources (e.g., permafrost soil) in both contemporary and paleo-
reconstructed carbon cycling of the Arctic (Karlsson et al., 2016; Tesi et al., 2016; Vonk et
al., 2012; Vonk et al., 2014).

With this overarching goal in mind, here we investigate the >10 μm fraction of

particulate organic matter (POM) in ice-covered and ice-free MIZ regimes of the Siberian
Arctic Shelf during the SWERUS-C3 expedition (July-August 2014) (Fig. 1). The plankton-
dominated POM samples collected throughout the ca. 4,500 km long cruise track were
characterized using bulk parameters (OC, $\delta^{13}C$ and $\Delta^{14}C$) and biomarkers (highly branched
isoprenoids, IP$_{25}$; CuO oxidation products). In addition, continuous measurements of
dissolved CO$_2$ (CO$_{2aq}$) and its stable carbon isotope composition ($\delta^{13}C_{CO2}$) were performed
during the campaign (Humborg et al., 2017) and used for a direct comparison with the
chemical composition of the POM fraction.

## 2. Study region

The Laptev Sea and the East Siberian Sea are shallow epicontinental seas in the Russian Arctic separated by the New Siberian Islands (Fig. 1). Sea-ice cover lasts for most part of the year over the shelf. Late spring/summer is characterized by the seasonal sea-ice retreat coupled with river freshet which supplies large amount of terrestrial carbon in the form of particulate and dissolved matters (Karlsson et al., 2016; Salvadó et al., 2016; Sánchez-García et al., 2011). The Lena (523 km3/y), Indigirka (54 km3/y), and Kolyma (48 km3/y) are the major rivers (Gordeev, 2006). During the ice-free season, the Lena plume can be traced in the outer-shelf of the Laptev Sea (Fichot et al., 2013; Salvadó et al., 2016; Sánchez-García et al., 2011) while Pacific inflow from the Bering strait affects further east the East Siberia margin (Semiletov et al., 2005). The Pacific inflow exerts control on the nutrient balance as it supplies phosphorous and silicate to an otherwise nutrient-depleted region (Anderson et al., 2011; Semiletov et al., 2005). Another important source of particulate material to the continental margin is the Pleistocene Ice Complex Deposit (ICD) entering the ocean via coastal erosion (Lantuit et al., 2011; Vonk et al., 2012) which is the dominant carbon source between the Kolyma river and the Lena river (Vonk et al., 2012).

## 3. Methods

### 3.1 POM (<10 μm) sampling

Seawater was pumped from a stainless steel inlet on the hull of the icebreaker *Oden* positioned at 8 m below the sea surface. The inlet system is tested and further described in Sobek and Gustafsson (2004) and Gustafsson et al. (2005). Figure 1a and 1b show the regions covered to harvest each POM (>10 μm) sample with their location shown as time-averaged position. The particulate material was retained via a large volume filtration apparatus using a 10-μm Nitex® (nylon) mesh placed in a 29.3 cm filter holder. After collection, filtered

particulate material was transferred in pre-clean HDPE tubes by rinsing the Nitex® filters
with MilliQ water. Samples were kept frozen throughout the expedition. In the lab, samples
were transferred in pre-cleaned Falcon® tubes (rinsed with 0.1M HCl) and gently centrifuged
to remove the supernatant. The residual particulate material was frozen and subsequently
freeze-dried prior to biogeochemical analyses.

3.2 Bulk carbon isotopes and biomarker analyses
Organic carbon (OC) and stable carbon isotope ($\delta^{13}$C) analyses were carried out on
acidified samples (Ag capsules, HCl, 1.5M) to remove the carbonate fraction (Nieuwenhuize
et al., 1994). Analyses were performed using a Thermo Electron mass spectrometer directly
coupled to a Carlo Erba NC2500 Elemental Analyzer via a Conflo III (Department of
Geological Sciences, Stockholm University). OC values are reported as weight percent
(%d.w.) whereas stable isotope data are reported in the conventional $\delta^{13}$C notation (‰). The
analytical error for $\delta^{13}$C was lower than ±0.1‰ based on replicates. Acidified (HCl, 1.5 M)
samples for radiocarbon abundance were analysed at the US-NSF National Ocean Science
Accelerator Mass Spectrometry (NOSAMS) facility (Woods Hole Oceanographic Institution,
Woods Hole, USA). Radiocarbon data are reported in the standard $\Delta^{14}$C notation (‰).
Alkaline CuO oxidations were carried out using an UltraWAVE Milestone microwave
as described in Tesi et al. (2014). Briefly, about 2 mg of OC was oxidised using CuO under
alkaline (2N NaOH) and oxygen-free conditions at 150 °C for 90 min in teflon tubes. After
the oxidation, known amounts of recovery standards (trans-cinnamic acid and ethylvanillin)
were added to the solution. The NaOH solutions were then acidified to pH 1 with
concentrated HCl and extracted with ethyl acetate. Extracts were dried and redissolved in
pyridine. CuO oxidation products were quantified by GC-MS in full scan mode (50-650 m/z).
Before GC analyses, the CuO oxidation products were derivatized with bis(trimethylsilyl)
trifluoroacetamide+1% trimethylchlorosilane at 60°C for 30 min. The compounds were
separated chromatographically in a 30m×250 μm DB5ms (0.25 μm thick film) capillary GC
column, using an initial temperature of 100°C, a temperature ramp of 4°C/min and a final
temperature of 300°C. Lignin phenols (terrestrial biomarkers) were quantified using the
response factors of commercially available standards (Sigma-Aldrich) whereas the rest of the
CuO oxidation products were quantified by comparing the response factor of trans-cinnamic
acid. Lignin-derived reaction products include vanillyl phenols (V=vanillin, acetovanillone,
vanillic acid), syringyl phenols (S=syringealdehyde, acetosyringone, syringic acid) and
cinnamyl phenols (C=p-coumaric acid, ferulic acid). In addition to lignin, cutin-derived
products (hydroxyl fatty acids) were used to trace the land-derived input (Goñi and Hedges,
1990; Tesi et al., 2010). Other CuO oxidation products include para-hydroxybenzene
monomers (P-series), benzoic acids (B-series) and short-chain fatty acids (FA-series) which
can have both terrestrial and marine origin (Goñi and Hedges, 1995; Tesi et al., 2010).
The sea-ice proxy $IP_{25}$ (mono-unsaturated highly branched isoprenoid (HBI) alkene)
was quantified according to Belt et al. (2012). $IP_{25}$ producers are a minor (<5%) fraction of
the total sea-ice taxa which are, however, ubiquitous in pan-Arctic sea-ice. Species include
*Pleurosigma stuxbergii var. rhomboide*, *Haslea crucigeroides* (and/or *Haslea spicula*) and
*Haslea kjellmanii* (Brown et al., 2014a). Briefly, lipids were extracted via sonication using a
dichloromethane/methanol solution (2:1 v/v × 3). Prior to the extraction, two internal
standards (7-hexylnonadecane, 7-HND and 9- octylheptadecene, 9-OHD) were added to
permit quantification of $IP_{25}$ (monounsaturated highly branched isoprenoid) following
analysis via GC-MS. Total lipid extracts (TLEs) were dried under $N_2$ after removing the water
excess with anhydrous $NaSO_4$. Dry TLEs were redissolved in dichloromethane and the non-
polar hydrocarbon fraction was purified using open column chromatography (deactivated
SiO$_2$) and hexane as eluent. Saturated and unsaturated n-alkanes were further separated using
10% AgNO$_3$ coated silica gel using hexane and dichloromethane, respectively.
Quantification of IP$_{25}$ was carried out in SIM mode (*m/z* 350.3) as described in Belt et
al. (2012). The GC was fitted with a 30m×250 μm DB5ms (0.25 μm thick film) capillary GC
column. Initial GC oven temperature was set to 60°C followed by a 10°C/min ramp until a
final temperature of 310°C (hold time 10 min).

**3.3. Microscope images of plankton**
High resolution digital images were taken with an Environmental Scanning Electron
Microscope (ESEM) Philips XL30 FEG in high voltage (15kV) and magnification 250X.
Samples were further studied for identification of diatoms and dinoflagellates using a
transmitted light microscope (Leitz Laborlux 12 Pol) equipped with differential interference
contrast optics at 1000X magnification. Microscope slides were prepared using settling
chambers to achieve an even distribution of particles on the cover glass, regardless of size and
shape (Warnock and Scherer, 2015).

**3.4 Sea-ice data**
Daily AMSR2 sea-ice extent and concentration maps were provided by the Institute of
Environmental Physics, University of Bremen, Germany (Spreen et al., 2008) as GeoTIFF
files (ftp://seaice.uni-bremen.de).

**3.5 Statistics**
We used two-tailed T-test (homoscedasticity) and Welch T-test (heteroskedasticity) to
assess whether the differences between open waters and sea-ice dominated waters were
statistically significant. For this study, significance level (alpha) was set at 0.01.

## 4. Surface water conditions during the SWERUS-C3 expedition

970  Before discussing the chemical composition of the POM (>10 μm), here we briefly

971 introduce the different environmental conditions encountered throughout the cruise track. The

972 surface water data presented in this section were pulled together from previous studies which

973 provide an in-depth analysis of the surface water properties during the SWERUS-C3

974 expedition in 2014 (Humborg et al., 2017; Salvadó et al., 2016) (Table 2). For this study,

975 continuous $p\mathrm{CO_2aq}$ and $\delta^{13}\mathrm{C_{CO2}}$ data (Humborg et al., 2017) were averaged to match the

976 water sampling stations allowing for a direct comparison with DOC and salinity data (Fig. 2)

977 (Supplementary Material).

978  Summer 2014 was consistent with the long-term downward trend in Arctic sea-ice

979 extent. The strongest anomalies were observed in the LS which experienced the most

980 northerly sea-ice shift since satellite observations began in 1979 (National Snow and Sea Ice

981 Data Center, NSIDC. http://nsidc.org/data). Unpublished data). In general, sea-ice displayed a

982 strong gradient over the study region going from ice-free conditions in the outer LS to ice-

983 dominated waters in the outer ESS (Fig.1.) Three snapshots of the sea-ice extent and

984 concentrations (.i.e. at the beginning, in the middle and at the end of the sampling) is shown in

985 Fig.1. Furthermore, Table 1 reports the averaged sea-ice concentrations encountered during

986 the collection of each sample.

987  The surface water salinity exhibited a longitudinal trend characterized by low values

988 in the outer LS while the sea-ice dominated ESS waters showed relatively higher values (Fig.

989 2a; Table 2). However, the highest salinity values were measured in the westernmost stations

990 resulting in a sharp gradient in the LS. The low surface water salinities in the outer LS are

991 most likely the result of both Lena river input and sea-ice thawing (Humborg et al., 2017) that

992 started in late May (Janout et al., 2016).

The highest DOC concentrations were measured in the mid-outer LS in the surface
water plume affected by Lena River runoff (Fig.2b; Table 2). Overall, DOC concentrations
followed the plume dispersion with high DOC concentrations corresponding to low salinities
(Fig. 2). Carbon stable isotopes ($\delta^{13}$C) and terrestrial biomarkers (of the solid-phase extracted
DOC fraction; Salvado et al., 2016) further confirmed the influence of terrestrial DOC in the
outer LS, while the land-derived input progressively decreased moving eastward.
$p$CO$_2$aq concentrations exhibited a typical estuarine pattern over the study region
(Humborg et al., 2017) (Fig. 2d; Table 2). Low salinity waters in the outer LS showed above
atmospheric CO$_2$ concentrations (i.e., supersaturated) while surface waters below sea-ice
exhibited undersaturated concentrations. The most depleted $\delta^{13}$C$_{CO2}$ values were measured off
the Lena river mouth (Fig. 2e; Table 2). Being relatively rich in land-derived material, it is
likely that respired terrestrial OC within the Lena river plume exerted control on the CO$_2$
isotopic signature and concentration (Humborg et al., 2017).
Finally, nutrient distribution revealed nitrate (NO$_3$) depletion in surface waters
throughout the cruise track (Humborg et al., 2017) in comparison with the Arctic Ocean
gateways such as the Bering strait. Here, nutrient concentrations in surface waters are two-
order of magnitude higher compared the our study region (Torres-Valdés et al., 2013).
Phosphate (PO$_4$) exhibited rather low concentrations in the outer LS and relatively higher
concentrations below the sea-ice in the outer ESS (Humborg et al., 2017) likely reflecting the
inflow of nutrient-rich Pacific waters (Anderson et al., 2011; Semiletov et al., 2005; Torres-
Valdés et al., 2013).

**5. Results and discussion**
**5.1 Source of the POM (>10 μm) fraction**
The Arctic Ocean off northern Siberia receives large quantities of dissolved and
particulate terrestrial organic carbon  via continental runoff and coastal erosion (Alling et al.,
2010; Dittmar and Kattner, 2003; McClelland et al., 2016; Sánchez-García et al., 2011;
Semiletov et al., 2013; Vonk et al., 2012). The land-derived material that does not settle in the
coastal zone further travels across the continental margin reaching out to the outer-shelf
region resuspended within the benthic nepheloid layer or in suspension within the surface
river plume (Fichot et al., 2013; Sánchez-García et al., 2011; Wegner et al., 2003). Another
fraction of terrestrial material can travel across the Siberian margin trapped in fast ice
(Dethleff, 2005). Considering the potential allochthonous contribution, we addressed to what
extent terrestrial organic material affects the POM (>10μm) fraction by quantifying the
concentration of lignin phenols and C16-18 hydroxy fatty acids (cutin-derived products).
These biomarkers are exclusively formed by terrestrial vegetation and, thus, serve as tracers
of land-derived material in the marine environment (Amon et al., 2012; Bröder et al., 2016b;
Feng et al., 2015).
Upon CuO alkaline oxidation the POM (>10μm) samples yielded only traces of lignin
phenols while the cutin-derived products were not detected (Fig. 3). Other oxidation products
in high abundance included saturated and mono-unsaturated short chain fatty acids (C12-
18FA), para-hydroxy phenols, benzoic acids and dicarboxylic acids. These other reaction
products are ubiquitous in both marine and terrestrial environments but they are predominant
in plankton-derived material, especially short-chain fatty acids (Goñi and Hedges, 1995).
When compared with active-layer permafrost soils and ice-complex deposits (Tesi et al.,
2014), POM (>10μm) samples displayed a distinct CuO fingerprint dominated by short chain
fatty acids (Fig. 3), consistent with the typical CuO products yields by phytoplankton batch
cultures upon CuO alkaline oxidation (Goñi and Hedges, 1995). SEM images further
corroborated the abundance of marine plankton detritus in the POM (>10μm) fraction while
lithogenic particles (clastic material) appeared to be sporadic in all samples.
The OC content (% d.w.) of the POM (>10um) fraction decreased eastwards showing
high concentrations in the LS and relatively low values in the ESS (Table 1; $p<0.01$ T-test).
However, in terms of absolute concentration in the water column (μC/l), the highest levels
were generally observed in the sea-ice covered region (Table 1; Fig. 4a; $p<0.01$ T-test).
Qualitative analyses by SEM and transmitted-light microscopy highlight important
differences in plankton assemblages which reflect different timing of the plankton blooms
which can explain these differences in concentration. Specifically, the open-water LS stations
exhibited a low degree of plankton diversity and were largely dominated by a bloom of
heterotrophic dinoflagellate cysts (*Protoperidinium* spp) (Fig. 5a; Table 3). Moving towards
the ice-dominated regions, diatoms become the prevailing species. Dominant diatom genera
include *Chaetoceros spp.* (dominant diatom in several stations), *Thalassiosira spp.*,
*Rhizosolenia spp.*, *Coscinodiscus spp.*, *Asteromphalus spp.*, *Navicula spp.* as well as sea-ice
species such as *Fragilariopsis cylindrus* and *Fragilariopsis oceanica* (Fig. 5b,c; Table 3).
Moored optical sensors deployed in the LS shelf recorded the sea-ice retreat in 2014
and found no sign of pelagic under-ice blooms despite available nutrients while high
chlorophyll concentrations were detected immediately after the ice retreated in late May
(Janout et al., 2016). The ice-edge blooms lasted for about 2 weeks according to the high
resolution chlorophyll time-series (Janout et al., 2016). Thus, our post-bloom sampling in the
LS essentially captured an oligotrophic environment dominated by heterotrophic
dinoflagellate cysts (i.e, Protoperidinium spp) which likely fed on phytodetritus and river-
derived organic material. Such conditions are fairly consistent with the relatively low carbon
contents (μgC/L) observed in LS waters (Fig. 4a).
The Arctic sea-ice biomarker IP25 (Fig. 4b) further highlights the different regimes
observed in ice-free and ice-dominated surface waters. IP25 is a proxy of sea-ice based on a
highly branched mono-unsaturated isoprenoid alkene found in some sea-ice diatoms which,
however, generally account for 5% of the total sea-ice taxa (Belt et al., 2007; Brown et al.,
2014b). The IP25 concentrations varied by several orders of magnitude over the study area
showing low concentrations in the open-water western region while the sea-ice dominated
surface waters to the east exhibited high concentrations especially at station 31b (Fig. 4b;
Table 1) ; $p < 0.01$ Welch T-test). The fact that IP25 was still detectable throughout the ice-free
outer LS suggests that the proxy captured the signal of the sea-ice retreat that occurred shortly
before the sampling at the end of May/early June (Janout et al., 2016). Alternatively, the IP25
could have been advected from nearby sea-ice dominated regions.

**5.2. Dual carbon isotopes: $\delta^{13}C$ and $\Delta^{14}C$**
$\delta^{13}C$ and $\Delta^{14}C$ of the POM ($>10\mu m$) samples exhibited a distinctive longitudinal trend
across the study area between LS and ESS (Fig. 4c,d) ($p < 0.01$ T-test) . Depleted $\delta^{13}C$ values
characterized the LS open waters ranging from -28.1 to -24.7‰ (Fig. 4c). Although within the
range of terrestrially-derived material, our CuO oxidation data (i.e. trace of lignin phenols and
absence of cutin-derived products) suggest that the "light" isotopic composition in the LS
might instead reflect the plankton assemblage dominated by heterotrophic dinoflagellate cysts
as previously described (e.g., *Protoperidinium* spp; Fig. 5a). More specifically, heterotrophic
dinoflagellates can adapt their metabolism depending on the substrate available (e.g., diatoms
and bacteria). Several studies have shown that terrestrial DOC greatly promotes bacteria
biomass production which in turn stimulates the growth of heterotrophic dinoflagellates
(Carlsson et al., 1995; Purina et al., 2004; Wikner and Andersson, 2012). Thus, in these
conditions, allochthonous terrestrial DOC is actively recycled by bacteria and transferred to
dinoflagellates which explains, thus, the depleted $\delta^{13}C$ values observed in the river-dominated
samples (Carlsson et al., 1995).

The modern radiocarbon fingerprint of the Lena DOC discharge is consistent with

$\Delta^{14}C$ signature of the POM ($>10\mu m$) fraction in the LS (up to +99 ‰), supporting the
importance of terrestrial DOC as a carbon source for the food web in the river plume (Fig. 4d
and 6). By contrast, comparison with other potential carbon sources which include the Lena
river particulate organic carbon, surface sediments, Pleistocene coastal Ice-Complex Deposit
and Pacific DIC inflow reveals a different (more depleted) radiocarbon fingerprint (Fig. 6). It
is also import to highlight that the DOC within the Lena plume is one/two-order of magnitude
higher than the particulate carbon pool supporting, thus, our hypothesis (Humborg et al.,
2017; Salvadó et al., 2016).

Moving towards the ice-dominated ESS, surface waters progressively became more

autotrophic and productive (Humborg et al., 2017) while the POM ($>10\mu m$) exhibited a wide
$\delta^{13}C$ signature ranging from -28.6 to -21.2‰ (Fig. 4c). The most depleted values were
observed across the transition zone between open-waters and sea-ice. Visual inspections of
these samples revealed large abundance of the centric diatom *Chaetoceros* spp. (spores and
vegetative cells; St22, Fig. 5b) while lignin and cutin data indicated, a negligible input of
land-derived material. Primary factors determining the fractionation of stable carbon isotopes
in phytoplankton are several and include $CO_2aq$ concentration, $\delta^{13}Caq$, growth rate, cell size,
cell shape, light and nutrient availability (Gervais and Riebesell, 2001; Laws et al., 1997b;
Popp et al., 1998; Rau et al., 1996). Our understanding about isotopic fractionation has been
historically achieved via laboratory experiments designed to test each factor under controlled
conditions. In natural environments, however, different factors can compete with each other,
sometimes in opposite directions. Yet, the existing knowledge about surface water properties
during the expedition (Humborg et al., 2017) can provide important constraints for the
isotopic signal interpretation.

For example, comparison with continuous $\delta^{13}C$-$CO_2$aq and $p$$CO_2$aq data measured

throughout the cruise track - time-averaged to match the large volume filtration along the
cruise track (Table 1) - suggested a negligible role exerted by $\delta^{13}C$-$CO_2$aq (Fig. 7b) while
$p$$CO_2$aq concentration correlated with the $\delta^{13}C$ of the POM (>10$\mu$m) fraction ($r^2$=0.72;
$p$<0.01) (Fig. 7a). Such a relationship fits with the general model according to which a low
demand (i.e., low growth rate) and high supply (i.e., abundant $CO_2$aq) favour high
fractionation and vice versa (Laws et al., 1997a; Laws et al., 1995; Wolf-Gladrow et al.,

1999).

During the expedition, surface water properties (i.e. $O_2$ and $CO_2$, Table 2) (Humborg

et al., 2017) suggest that the productivity in the outer ESS increases moving eastward, as
commonly observed, likely due to the Pacific inflow (Anderson et al., 2011; Semiletov et al.,
2005). As a result, the wide range of plankton $\delta^{13}C$ over the ESS can be explained in terms of
two different regimes: (a) in the transition zone between open waters and sea-ice, the
productivity was low but $p$$CO_2$aq was supersaturated while (b) in the easternmost ESS,
productivity was high but $p$$CO_2$aq was depleted (Fig. 7b). The former regime favours
fractionation while the latter does not (Fig. 7b). Different diatom assemblages can also be
another factor to consider although the phytoplankton diversity observed over ESS can be
considered rather small (e.g. *Chaetoceros spp.* dominant in most of the samples) compared to
the wide range of $\delta^{13}C$ observed (i.e., from -28.8 to -21.6) (Table 3).

The POM (>10$\mu$m) fraction in the sea-ice dominated ESS exhibited slightly - but

consistently - depleted $\Delta^{14}C$ values ranging from -62 to -49 ‰ (Fig. 4d). This region is
affected by the inflow of Pacific waters whose DIC exhibits, however, a modern $\Delta^{14}C$
signature (Griffith et al., 2012) (Fig. 6). By contrast, these results suggest the influence from
an aged carbon pool. As the ESS remains covered by sea-ice for most of the year, it is
possible that the sea-ice hampers the gas exchange with the atmosphere and acts as a lid by
trapping $CO_2$ which derives from the breakdown of sedimentary organic material (Anderson
et al., 2009; Semiletov et al., 2016), which might have such ages (Bröder et al., 2016a; Vonk
et al., 2012). In these conditions, the pre-aged $CO_2$ accumulates underneath the sea-ice and is
subsequently incorporated during carbon fixation by the phytoplankton. While supersaturated
bottom waters were extensively documented in the region with important consequences on the
local DIC (Anderson et al., 2009; Pipko et al., 2009), more work is clearly needed to
understand if early diagenesis in sediments can also affect the radiocarbon signature of the
$CO_2$aq underneath the sea-ice. Alternatively, the slightly depleted radiocarbon signature might
indicate the presence of pre-aged terrestrial organic carbon (Fig. 6) in the POM (>10μm)
samples, not reflected in the lignin and cutin tracers (Fig. 3). However, it would then remain
elusive why such an aged land-derived influence was not visible in the river-dominated LS
waters while it affected the sea-ice dominated region.
Taken together, our results indicate that the dual-carbon isotope fingerprint is highly
affected by the trophic conditions (heterotrophic *vs* autotrophic) as well as the extent of
primary productivity. In a warming scenario characterized by sea-ice retreat (Arrigo et al.,
2008; Comiso et al., 2008) and enhanced terrestrial input from land as result of hydrology and
permafrost destabilization (Frey and Smith, 2005; Vonk et al., 2012), the geochemical
composition of plankton will likely change as the warming proceeds.

**6. Conclusions**
Analyses of large-volume filtrations of plankton-dominated >10 μm particle samples
revealed a high degree of heterogeneity in the dual carbon isotope signature ($\delta^{13}C$ and $\Delta^{14}C$)
between ice-free waters (Laptev Sea) and the ice-covered region (East Siberian Sea).
Our results suggest a heterotrophic environment in the outer LS open waters where the
$\delta^{13}C$ depleted river DOC is transferred to relatively higher trophic levels via microbial
incorporation in the river plume. Moving eastwards towards the ice-dominated outer ESS,
surface waters became progressively more autotrophic. Here, the isotopic fractionation
appears to follow the phytoplankton growth *vs* $CO_2$ demand model according to which carbon
fractionation decreases at high growth and low $CO_2$ concentrations. As a result, the transition
between open-waters and sea-ice exhibited more depleted $\delta^{13}C$ values compared to the
productive easternmost stations. Radiocarbon signatures were slightly depleted over the whole
sea-ice dominated area. This raises the question whether the sea-ice hampers the gas exchange
with the atmosphere and trap the $CO_2$ sourced from reactive sedimentary carbon pools.
In a warming scenario, it is likely that the oligotrophic ice-free LS will be dominated
by heterotrophic metabolism fuelled by terrestrially-derived organic material (i.e., Lena
input). In these conditions, the dual-carbon isotope signature of the heterotrophic plankton
will essentially reflect the terrestrial fingerprint. In the ESS, which receives the inflow of the
nutrient-rich Pacific waters, ice-free conditions will enhance light penetration. This in turn
might further stimulate phytoplankton growth with important implications in terms of $CO_2$
depletion and resulting low isotope fractionation. Altogether, this will result in a sharp
compositional gradient (e.g. $\delta^{13}C$) between LS and ESS similar to what captured in our semi-
synoptic study.

**Acknowledgements**
We thank the *I/B Oden* crew and the Swedish Polar Research Secretariat staff. This
study was supported by the Knut and Alice Wallenberg Foundation (KAW contract
2011.0027), the Swedish Research Council (VR contract 621-2007-4631 and 621-2013-
5297), European Research Council (ERC-AdG CC-TOP project #695331 to Ö.G.). T. Tesi

additionally acknowledges EU financial support as Marie Curie fellow (contract no. PIEF-GA-2011-300259). J.A. Salvadó acknowledges EU financial support as a Marie Curie grant (contract no. FP7-PEOPLE-2012-IEF; project 328049). I. Semiletov acknowledges financial support from the Russian Government (grant No. 14, Z50.31.0012/03.19.2014) and the Russian Foundation for Basic Research (nos. 13-05-12028 and 13-05-12041), and E. Panova from the Russian Scientific Foundation (grant no. 15-17-20032). We thank the Arctic Great Rivers Observatory (NSF-1107774) for providing DOC and POC river data (www.arcticgreatrivers.org). This is ISMAR publication ID n.1940.

**Table 1. Chemical composition of the POM (>10μm) fraction and continuous CO₂aq measurements***

| ID | Time averaged latitude (N) | Time averaged longitude (E) | Mean sea-ice percentage (%) | POM (>10μm) concentration (mg/l) | OC (d.w.) | $\delta^{13}C$ (‰) | $\Delta^{14}C$ (‰) | IP25 (ng/gOC) | averege CO₂aq (ppm)* | average $\delta^{13}C$-CO₂aq (‰)* |
|---|---|---|---|---|---|---|---|---|---|---|
| ST4 | 81.68 | 105.96 | 98.4 | 6 | 18.2 | -26.7 | n.d. | n.d. | 323 | -10.9 |
| ST5 | 80.47 | 114.07 | 98.7 | 15 | 42.6 | -27.6 | n.d. | n.d. | 322 | -11.0 |
| ST6 | 78.86 | 125.22 | 82.2 | 1 | 51.7 | -26.6 | 99 | n.d. | 325 | -10.8 |
| ST7 | 77.88 | 126.62 | 0.0 | 11 | 43.1 | -25.7 | n.d. | 88 | 350 | -10.7 |
| ST8 | 77.16 | 127.32 | 0.0 | 17 | 30.9 | -26.7 | 41 | n.d. | 391 | -10.5 |
| ST9 | 76.78 | 125.83 | 0.0 | 3 | 31.5 | -27.9 | 30 | 48 | 385 | -10.5 |
| ST10 | 76.90 | 127.81 | 0.0 | 11 | 40.9 | -24.7 | n.d. | n.d. | 349 | -11.0 |
| ST11 | 77.12 | 126.66 | 0.0 | 13 | 29.6 | -28.1 | 27 | 13 | 428 | -10.7 |
| ST22 | 77.67 | 144.63 | 0.0 | 20 | 11.3 | -28.8 | n.d. | 95 | 394 | -11.0 |
| ST23 | 76.43 | 147.53 | 0.0 | 6 | 7.6 | -28.5 | -50 | n.d. | 394 | -11.2 |
| ST24 | 76.42 | 149.84 | 34.4 | 19 | 11.9 | -26.8 | -62 | 368 | 374 | -11.1 |
| ST25 | 76.62 | 152.03 | 96.7 | 23 | 19.5 | -25.7 | -31 | 465 | 263 | -10.8 |
| ST26 | 76.14 | 157.85 | 96.2 | 109 | 30.8 | -24.2 | -30 | 217 | 316 | -10.9 |
| ST27 | 75.00 | 161.03 | 91.5 | 41 | 23.3 | -23.0 | n.d. | 256 | 299 | -11.1 |
| ST28 | 74.63 | 161.98 | 86.3 | 28 | 15.5 | -23.8 | n.d. | n.d. | 214 | -11.3 |
| ST29 | 73.61 | 169.72 | 79.3 | 31 | 14.7 | -23.2 | -50 | 518 | 184 | -11.3 |
| ST30 | 75.61 | 174.01 | 66.7 | 43 | 22.6 | -27.0 | n.d. | n.d. | 304 | -10.5 |
| ST31A | 75.85 | 174.41 | 75.6 | 30 | 10.9 | -21.6 | -62 | 1911 | 182 | -10.6 |
| ST31B | 74.26 | 173.74 | 63.5 | 15 | 4.6 | -23.3 | n.d. | 783 | n.d. | n.d. |
| ST32 | 73.56 | 176.06 | 51.8 | 21 | 11.3 | -24.5 | -58 | 131 | n.d. | n.d. |
| ST33 | 72.35 | -175.14 | 0.0 | 20 | 15.5 | -23.5 | n.d. | 473 | n.d. | n.d. |
| ST34 | 73.28 | -173.05 | 28.7 | 76 | 13.4 | -21.6 | -52 | 970 | n.d. | n.d. |
| ST35 | 75.21 | -172.05 | 53.9 | 24 | 14.3 | -24.2 | n.d. | 268 | n.d. | n.d. |

n.d = not determined
*Humborg et al. (2017)











**Table 2. Surface water (0-20 m) chemical and physical properties during the SWERUS-C3 expedition\***

| | Salinity | Temperature | DIC | DOC | POC | $\delta^{13}$C-DIC | NO$_2$-NO$_3$ | PO$_4$ |
|---|---|---|---|---|---|---|---|---|
| | | °C | μmol kg$^{-1}$ | μmol kg$^{-1}$ | μmol kg$^{-1}$ | ‰ | μmol kg$^{-1}$ | μmol kg$^{-1}$ |
| | median | median | median | median | median | median | median | median |
| Outer LS shelf (0-20 m) | 32.87 | 3.84 | 2139 | 149.1 | 7.9 | 0.75 | 0.21 | 0.27 |
| LS shelf break (0-20 m) | 33.56 | 0.57 | 2114 | 91.5 | 10.1 | 1.10 | 0.26 | 0.15 |
| Outer ESS shelf (0-20 m) | 29.45 | -1.33 | 1969 | 84.2 | 10.7 | 1.14 | 0.25 | 0.97 |
| ESS shelf break (0-20 m) | 28.23 | -1.32 | 1979 | 73.7 | 4.6 | 1.47 | 0.11 | 0.59 |
| | mean | mean | mean | mean | mean | mean | mean | mean |
| Outer LS shelf (0-20 m) | 31.17 | 3.40 | 2119 | 179.8 | 7.9 | 0.58 | 0.60 | 0.29 |
| LS shelf break (0-20 m) | 33.42 | 0.96 | 2111 | 97.5 | 10.0 | 1.10 | 0.61 | 0.16 |
| Outer ESS shelf (0-20 m) | 28.95 | -0.05 | 1949 | 95.8 | 11.9 | 1.26 | 0.26 | 0.95 |
| ESS shelf break (0-20 m) | 28.27 | -1.31 | 1975 | 72.0 | 4.6 | 1.49 | 0.12 | 0.60 |
| | s.d. | s.d. | s.d. | s.d. | s.d. | s.d. | s.d. | s.d. |
| Outer LS shelf (0-20 m) | 3.22 | 2.38 | 89 | 66.3 | 1.7 | 0.50 | 0.91 | 0.11 |
| LS shelf break (0-20 m) | 0.70 | 2.07 | 23 | 21.2 | 1.7 | 0.11 | 0.74 | 0.06 |
| Outer ESS shelf (0-20 m) | 1.41 | 2.28 | 75 | 30.2 | 4.6 | 0.49 | 0.12 | 0.19 |
| ESS shelf break (0-20 m) | 0.53 | 0.04 | 49 | 3.2 | 0.3 | 0.08 | 0.03 | 0.02 |

*data from Humborg et al. (2017) and Salvadó et al. (2016)















**Table 3. Qualitative plankton characterization of selected POM (>10μm) samples**

| ID | Region | Diatoms | Dinoflagellates | Other species |
|---|---|---|---|---|
| ST6 | LS | Few *Coscinodiscus* | None observed | |
| ST9 | LS | None observed | Few *Protoperidinium* | |
| ST11 | LS | None observed | Abundant *Protoperidinium* | |
| ST22 | LS-ESS | Abundant *Chaetoceros*, few *Rhizosolenia*, *Thalassiosira* | None observed | |
| ST25 | LS-ESS | High diversity. Abundant *Chaetoceros, few Rhizosolenia, Coscinodiscus, Thallasiosira, Asteromphalus, Navicula* | None observed | Silicoflagellate |
| ST31A | ESS | High diversity. Abundant *Chaetoceros, few Rhizosolenia, Thallasiosira, Bacterosira, Navicula* | None observed | |
| ST31B | ESS | High diversity. Few *Chaetoceros, Thallasiosira, Fragilariopsis* | Few *Protoperidinium* | |
| ST34 | ESS | Abundant *Chaetoceros*, few *Thalassiosira, Navicula* | Few *Protoperidinium* | |


















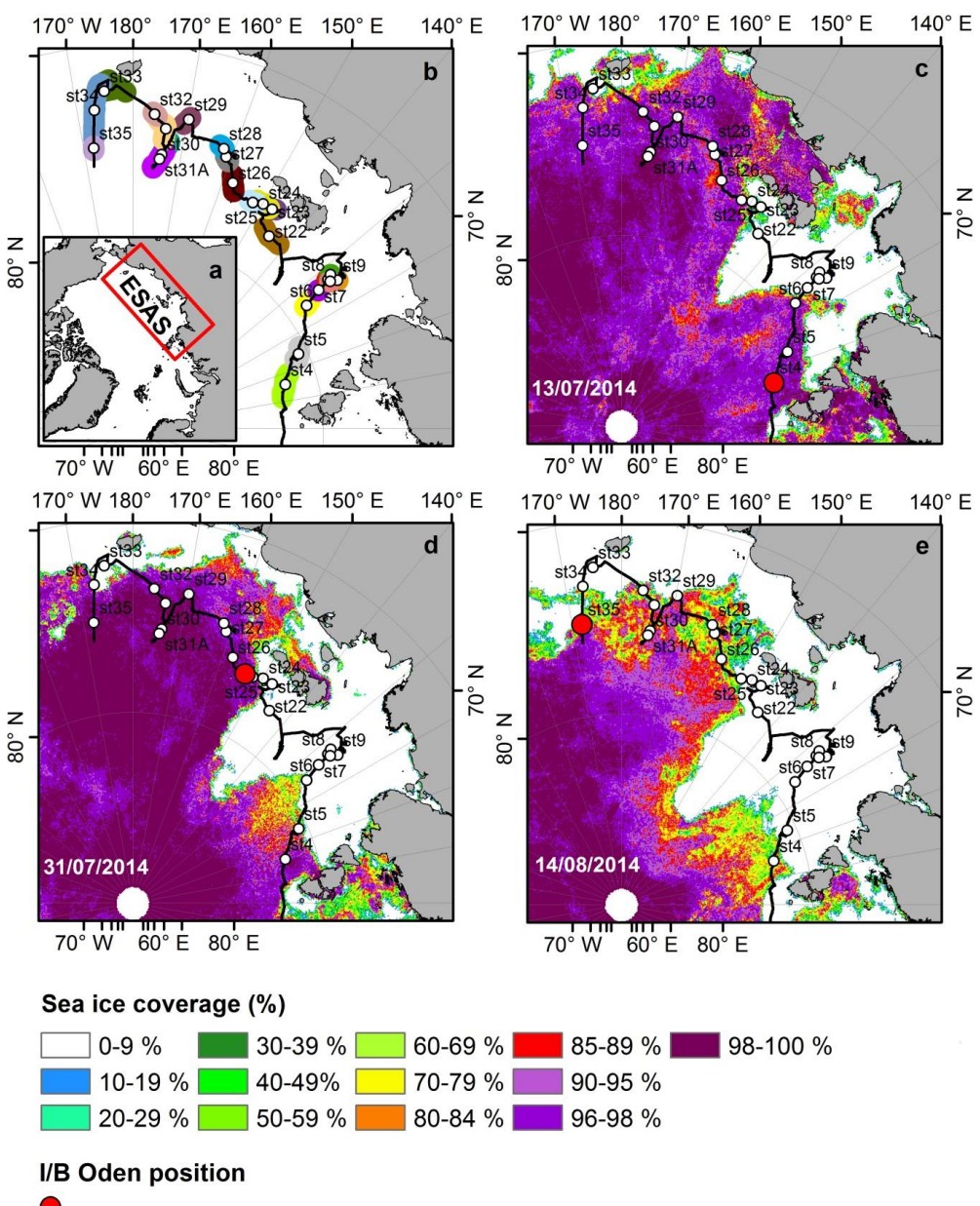


**Fig. 1** (a) The study area in the East Siberian Arctic Shelf. (b) Time-averaged position during the large-volume filtration (circles) of the POM (>10μm) samples. Shaded coloured areas show the sampling area covered to harvest each POM (>10μm) sample. Sea-ice extent and concentration at the beginning (c), in the middle (d) and at the end (e) of the sampling campaign. The ship position is shown by a filled red circle.

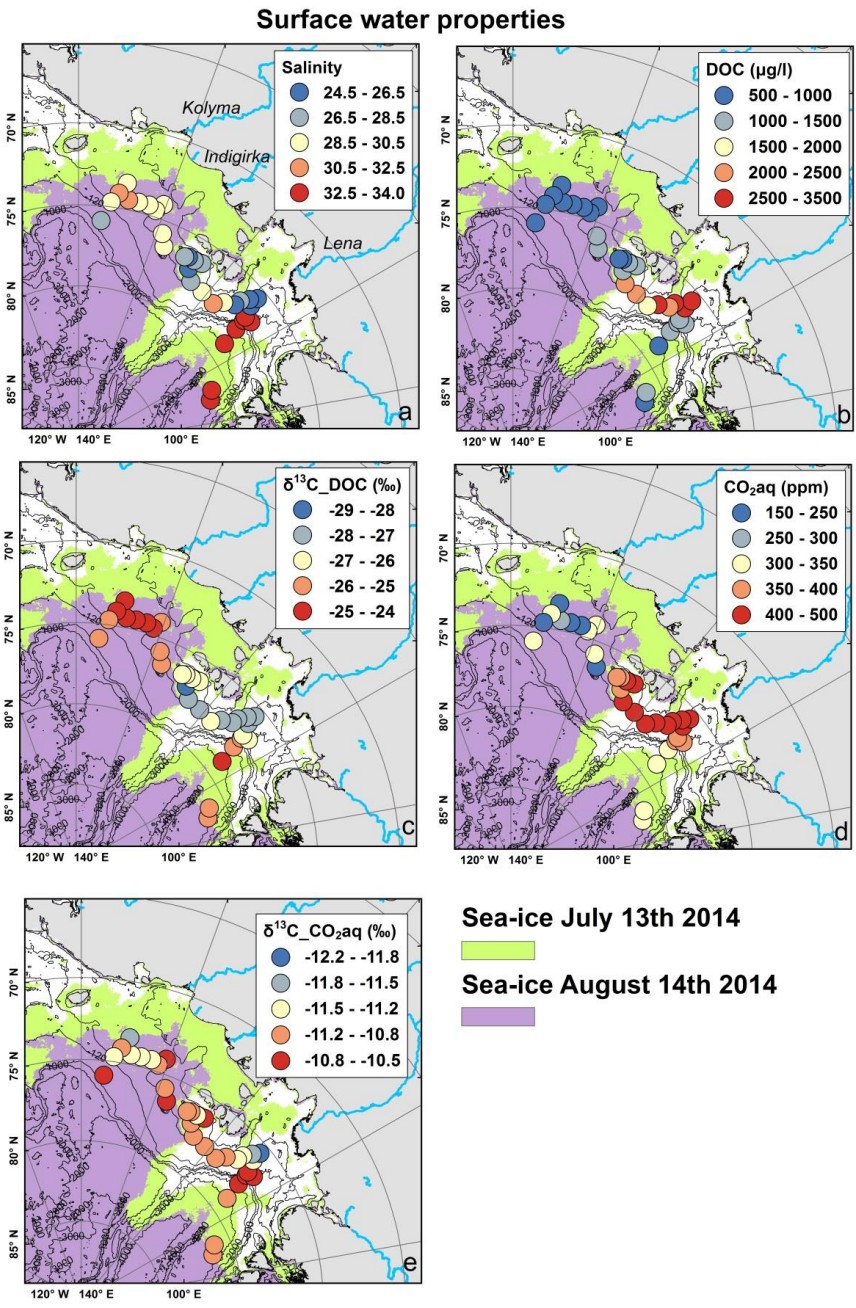

**Fig.2** Surface water properties. (a) Salinity. (b) DOC. (c) $\delta^{13}$C-DOC. (d) $CO_2$aq. (e) $\delta^{13}$C-$CO_2$aq. Shaded areas show the sea-ice extent at the beginning (13/07/2014) and at the end of the sampling campaign (14/08/2014) (Humborg et al., 2017; Salvadó et al., 2016).

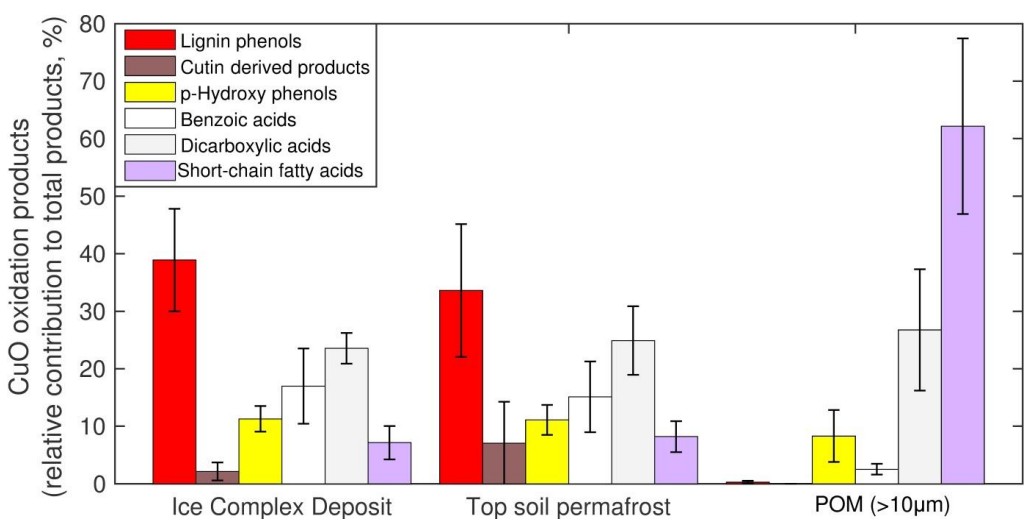

**Fig.3** Alkaline CuO fingerprint of top-soil permafrost samples (Tesi et al., 2014), Pleistocene Ice Complex Deposit (Tesi et al., 2014) and POM (>10μm) fraction (this study). The plot displays the relative proportion products yield upon alkaline CuO oxidation. The error bar refers to the natural variability of each dataset

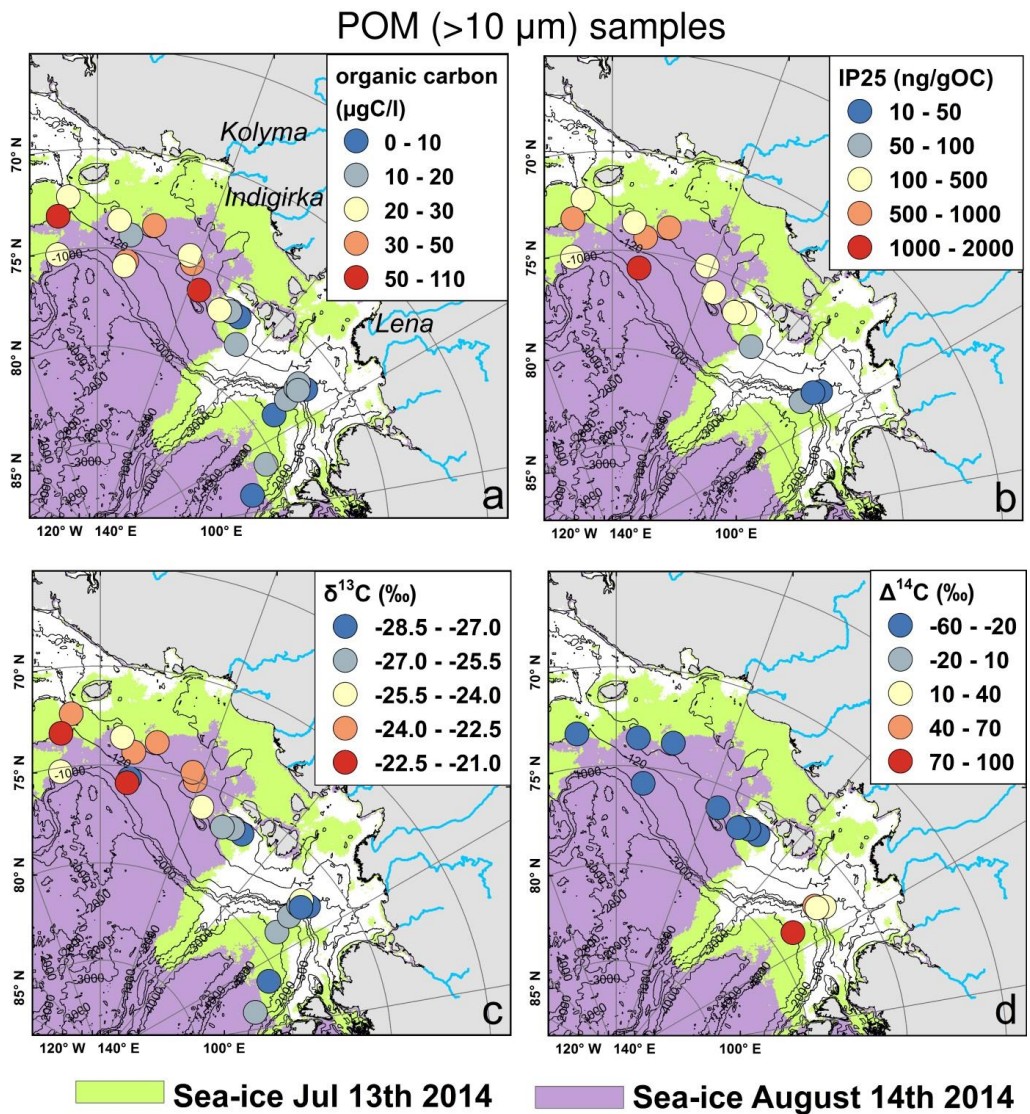

**Fig. 4** POM (>10μm) composition (a) Organic carbon concentration. (b) IP25 (mono-unsaturated highly branched isoprenoid. (c) $\delta^{13}$C. (d) $\Delta^{14}$C. Shaded areas show the sea-ice extent at the beginning (13/07/2014) and at the end of the sampling campaign (14/08/2014).

ST11 - Laptev Sea

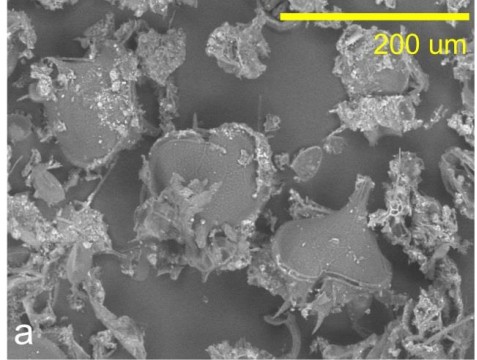

ST22 - Laptev Sea / East Siberian Sea

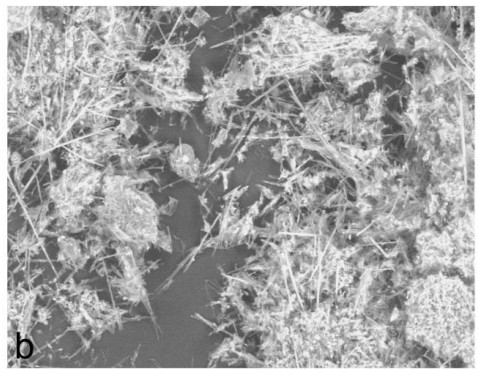

ST34 - East Siberian Sea

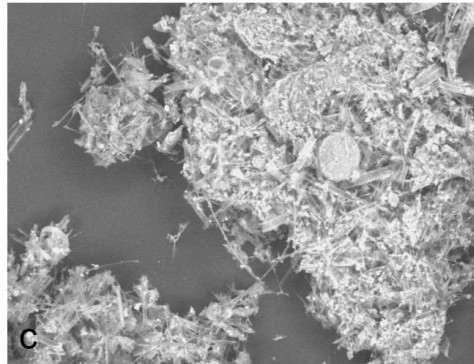



**Fig. 5** SEM images. (a) ST-11: Dinoflagellates (*Protoperidinium* spp.) in open-waters of the
Laptev Sea. (b) ST22: Diatoms, mostly spines (setae) of *Chaetoceros* spp. in the transition
between Laptev Sea and East Siberian Sea. (c) ST-34: Diatoms from sea-ice dominated
waters in the East Siberian Sea

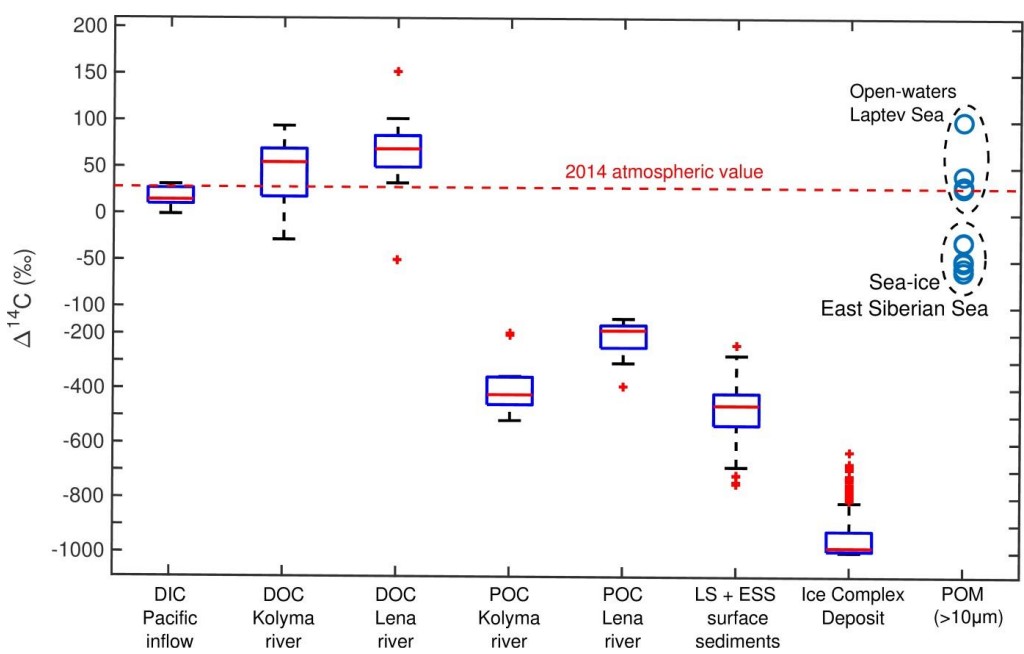




**Fig. 6** Radiocarbon signature of inorganic and organic carbon pools. Whisker plots of
radiocarbon values for different inorganic and organic carbon sources from the literature,
compared to the outer Laptev Sea and outer East Siberian Sea (blue circles, this study). Solid
lines show the median, the box limits display the $25^{th}$ and $75^{th}$ percentiles while the crosses
show the outliers. Source: DIC (Griffith et al., 2012), DOC-Kolyma (2009-2014), DOC-Lena
(2009-2014), POC-Kolyma (2009-2011), POC-Lena (2009-2011)
(www.arcticgreatrivers.org), Laptev Sea and Eastern Siberia Sea surface sediments (Salvadó
et al., 2016; Vonk et al., 2012) and Ice Complex Deposit (Vonk et al., 2012).





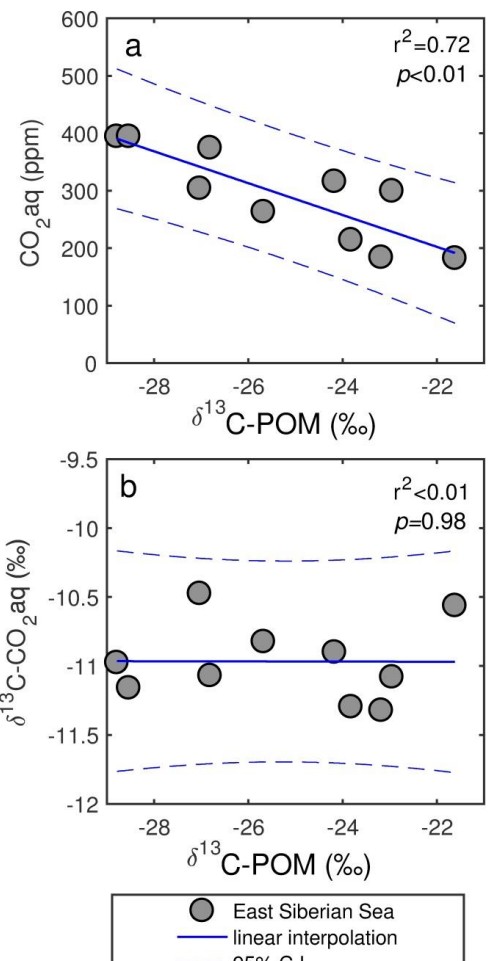




**Fig. 7** Correlations (a) $p$CO$_2$aq vs $\delta^{13}$C (POM (>10μm) fraction) and (b) $\delta^{13}$C-CO$_2$aq vs $\delta^{13}$C
in the East Siberian Sea (filled circles). The solid line shows the linear interpolation while the
dashed line shows the 95% confidence intervals.

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
