# Peer review of "Carbon geochemistry of plankton-dominated samples in the Laptev and East Siberian"

_Ocean Science, 2017_

## Author Comment (AC1) · 11 Apr 2017

Misspelled coauthor first name:

"Christoph Pearce" should be "Christof Pearce" instead

"Christophe Humborg" should be "Christoph Humborg" instead
* * *

---

## Referee Comment (RC1) · Anonymous Referee #1 · 27 Apr 2017

**Carbon geochemistry of plankton-dominated supra-1 micron samples in the Laptev and East Siberian shelves: contrasts in suspended particle composition.**

**General comments**

This study want to improve the understanding of the chemical composition of plankton that dominates regions of the Arctic Ocean characterized by different sea-ice coverages and and in the ice-free Laptev Sea. The authors conclude that terrestrial carbon influence the POM in the Laptev Sea with higher influence of the River Lena. In the East Siberian Sea with ice-cover the influence of land is smaller. This is a valuable study in an important part of the ocean with a paucity of observations.

The methodology for $^{13}$C and $^{14}$C analyses seem adequate. However, this is not the case for the estimates of plankton diversity based on qualitative data from scanning electron microscopy. Either the authors have to convince us that this method is adequate and provide quantitative data analyzed by proper statistical tests. Otherwise markedly constrain conclusions regarding phytoplankton diversity or remove this entirely, perhaps using references to other studies instead.

Also the $^{13}$C and $^{14}$C analyses consistently lack objective statistical tests to support the conclusions made. Even if this data in general looks convincing this needs to be added.

The text is quite OK but there are some sentences which are not currently understandable. It does not appear that the last version has been checked by an English speaking person, which is recommended.

Provided that these remarks and important specific comments below are remedied I recommend that the manuscript is published after a major revision.

**Specific comments**

Title: Please revise the title replacing the "supra1-micron samples" term with "POM".

r.28-36 I suggest to make the introductory paragraph shorter motivating the addressed question and the overall design of the study.

r.31-32 Unclear why "supra-" is used at this stage. Not a common term in my mind. Please just state "..the larger than 10 μm particulate organic matter (POM) fraction..." in the abstract. It´s clear from your definition what fraction that is covering.

r. 37-41 These conclusions need to be better supported by statistical tests as specified in the result and discussion section.

r. 42 "..communities via microorganisms". There are not reason to indicate several "loops". Please write concise and avoid unnecessary terms for clarity.

r. 44 The methodology does not seem adequate to assess the changes in diversity. Se comment in the result and discussion section. Also unclear what you mean by "…which is confirmed". Please rephrase.

r. 45-46 Unclear what is meant by "…follows the
general growth vs CO2aq supply model…". Please present in a clearer way.

r-48-50. What basis is this prediction based on? Please add a specification.

r. 65 What uptake are you referring to? Please rephrase and clarify.

r. 67 "…also project to water-…" Does not seem like proper English. Not understandable. Please rephrase.

r. 92-93 if the term supra- is not earlier defined and internationally agreed upon I see no reason to use it here. Just use POM and define the size fraction studies for simplicity.

r. 96. "..characterized by bulk…" If the MS has not been language checked by English speaking persons or dedicated companies please do so.

r. 92-101. This paragraph should be moved to the methods section.

r. 106 Please start with presenting the studied Sea areas and their characteristics. Please consider to use a map with sampling sites.

r. 112 Pleas add how the Falcon tubes were cleaned.

r. 119 How long time after sampling was the analysis? Was samples frozen all the time to analysis?

r. 166 How was samples taken and preserved? Were they concentrated in some way? Please add. Is there any reason why not other autotrophs like flagellates, picoplankton and cyanobacteria were included?

r. 170 What was the number of cells counted and precision of counting per sample?

r. 184 Please add the accuracy and precision of the measurements of $CO_2$ and $\delta^{13} C_{CO2}$.

r. 229 Please present (for all variables) some confidence intervals or test, validating what are statistically significant differences between stations (i.e. accounting for spatial and short term temporal variability). E.g. if you want to claim differences between sea areas show by a proper statistical methods that they are different from each other.

r. 236 The data referred to in Humborg et al. need either to be published before accepting this paper, or data presented in this paper.

r. 243 How do you define depletion? Please be more specific and refer to comparative data or references. Similar for "low" at r. 245.

r. 255 Please specify what "margin" that is referred to. This sentence is not possible to understand. Please rephrase.

r. 259 It´s not obvious how the concentration of lignin or hydroxyl fatty acids will say anything about effects on the POM fraction. Do you mean the conc. of these compounds in the POM? What about many other effects on living POM like species composition and functionality like growth or edibility? Do assume that most POM is non-living? Please motivate you analyses better relative the aim.

Fig.3 Please add what error bars are showing. Please specify what the values are relative against (carbon or mass?).

r. 276 Support you statement with a tests showing that these are different. What do a base the "high" and "low" assessments on (relative what)?

r. 275-287 SEM is not a proper method to asses diversity of phytoplankton (or present quantitative SEM data from sufficient number of samples?). That the diversity of phytoplankton is different between sea areas is therefore not sufficiently well demonstrated. Concentrations of at least major taxonomic groups is requested based on microscopy counting with adequate methodology (e.g. sediment chambers and reverse light microscopy) and statistical precision presented. Preferably also including flagellates, picoeucaryotes and cyanobacteria. Established diversity index should be used and tested for difference.

r- 288-289 Unclear how a line can detect a bloom. Please be more specific.

r.293-294 I don´t agree that the presented data convincingly show that dinoflagellates were dominating. How is the SEM preparation influencing different phytoplankton species? Is there a selections for robust dinoflagellate shells? Provide a reference or control experiment clarifying this.

r. 297-301 As presented isn´t IP25 then specifically indicating presence of sea-ice diatoms, not "..sampling of different plankton taxa…? Also can other sources of IP25 have contributed to the variability. Consider a re-interpretation. I suggest to calculate if the found conc. of IP25 could come from an expected conc. of diatoms in the sea ice and present that.

r. 309-310 Please provide a statistical test showing a significant trend of $CO_2$ and $\delta^{13} C_{CO2}$.

r. 316-317. On what basis is it assumed that the present dinoflagellates are hetero- or mixotrophic? That some dinoflagellates can eat bacteria is well shown in the literature. However, not that they are significant consumers of diatoms? Please provide a reference for this if so.

r. 324-325 I suggest to rephrase to "…, supporting the importance of terrestrial DOC as a carbon source for the food web in the river plume…. ).

Fig. 6 Pleas consider different symbols for the LS and ESS data for the observed POM (be consistent, POC used here?) $\Delta^{14}C$, instead of dashed circles. The presentation of sources and references can be moved to the methods section.

r. 341 Pleas provide a reference for opposite directions of $\delta^{13} C_{CO2}$ or make clear that this is a hypothesis.

Fig. 7 Be consistent with the use of POM (legend text) and POC (x-axis title).

r. 360-361 As stated above the presented SEM data does not properly account for phytoplankton diversity. Convincing tests and reference values showing that diversity is low is lacking.

r. 381-383 As you have not presented measures of heterotrophic or autotrophic (e.g. primary production) the sentence should be rephrased. Your $\delta^{13} C_{CO2}$ and other data rather show that trophic balance vary between the Sea area studied indicating larger terrestrial influence and importance of heterotrophic activity closer to river discharge, than under ice-cover (i.e. ESS)? I suggest to not directly refer to primary production (first time mentioned in the MS?) as no such direct analyses has been done.

r. 390-391 Something wrong with the end of the sentence. Please correct.

r. 393-394 As motivated above I don´t think it´s demonstrated convincingly that dinoflagellates dominate the phytoplankton community. Provide better quantitative date on diversity or remove the conclusion.

r.396 Please delete the parenthesis. It´s not correct to use the "microbial loop" concept as this primarily was used to imply a sink for carbon. In the same sentence you rather state that it's a link to higher trophic levels. The microorganisms is an integrated part of the marine food web globally, and in fact the original part of the food web in an evolutionary perspective. I have since long avoided the use of the term "microbial loop".

r. 397-404 As commented above you should provide statistical test for these conclusions. Either show that two sea areas are different from each other using stations as replicates, or show a significant trend (systematic pattern) in the whole transect supporting your conclusions.

**Technical comments**

To simplify for the reviewer and save time I prefer not to make further categorization of the comments, which may anyway be a matter of subjectivity. They are found above.

---

## Referee Comment (RC2) · Anonymous Referee #2 · 15 May 2017

General comments

The objective of this study was to investigate the composition of the suspended particulate organic matter in ice-covered and ice-free waters over the Laptev and East Siberian shelves. The main problem of this study is to assume that these samples are plankton-dominated, as indicated by the title. There are no data to support the fact that phytoplankton dominated the suspended particulate matter and such a dominance would actually be quite surprising over the shallow Siberian shelves (a lot of particulate material is resuspended and/or transported with the ice). An effort should be

made to quantify the phytoplankton contribution and composition before resubmitting this manuscript. If the authors somehow collected ice samples during this expedition, it would be relevant to compare the composition of the particulate matter in the ice with the composition of the suspended matter.

Another important problem is that too much of the current manuscript is based on another paper submitted elsewhere by many of the same authors that seems to be very similar to the current manuscript. This problem must be addressed.

Overall, while the study had the potential to provide interesting results from a very rarely sampled region, the current results do not bring very interesting or new information. It is well-known that ice covered regions are productive and display high concentrations of particulate matter. The interpretation of the results must be reevaluated in this context. Also, please keep in mind and specify throughout the manuscript that these are late summer observations and that conditions may be quite different during the productive spring period. Finally, the manuscript is too long, often repetitive, and the text needs to be revised by a native English speaker.

Specific comments

Title

I have never heard the terms supra-micron or supra-POM and I don't think there is a need for it. Please remove the term supra- throughout the manuscript.

Abstract

Lines 50-51: Comments like these are not informative. Always be specific.

Introduction

Lines 54-55: Provide more recent references.

Material and methods

Some information on the dates of sampling are required.

Lines 110-113: Several steps are unclear. When it is mentioned that particulate material was kept frozen, it means the filters? In which state were the samples transferred into the centrifuge? Were the samples thawed first?

Lines 115-116: Such information belongs in figure captions.

Lines 150 and 154: IP25 is a highly branched isoprenoid mono-unsaturated alkene. Introduce it properly and only once.

Section 2.3 Microscopic images of plankton* This is probably the biggest shortcoming of the study. It is baffling that the authors use microscopic images as a qualitative tool but did not include a quantification of the different phytoplankton groups. This would definitely improve the quality of the study. *Always precise if it is phytoplankton or zooplankton. Plankton is not a term precise enough.

The material and methods section is too long and often repetitive. Reduce.

Results/discussion

Section 3.1. Surface water conditions Most of this section does not belong in the paper. All the results (salinity, temperature, nutrients...) for which material and methods were not presented in the precedent section must be removed from the manuscript and the figures/tables as well. This is even more crucial considering that the same results are part of another submitted paper from the same authors. It is not appropriate to submit the same results twice and all the results that were submitted in Humborg et al. must be removed. Instead the authors should refer to these results in the discussion, which would be much stronger if or once the other paper is published. It would be more appropriate to start the discussion with section 3.2.

Lines 220-222: You should never write sentences in this form: Figure 1 displays. . . Table 1 reports. . . This is the type of mistakes made at the undergraduate level.

Line 225 and others: All maps (figures 1, 2 and 4) should be switched with North towards the top to help with the description of the results. This is the usual and correct way to place a map and it is less confusing when looking for the westernmost stations.

Lines 230-232: DOC concentrations mirrored. . . 'Mirrored' does not mean the opposite, it means similar. Get an English speaker to review your paper. And please limit your use of the word 'thus'.

Line 252: It is late and unnecessary to introduce the term TerrOC at this point. Either you introduce it earlier or you use other terms for consistency.

Lines 284-287: These phytoplankton species are typically observed late in the season. This should be specified. Chaetoceros and Thalassiosira are pelagic species growing in water only while Fragilariopsis cylindrus and oceanica grow both in ice and water (they are not sea ice species necessarily). More information could be obtained through extensive and quantitative taxonomic analyses of the existing samples.

Lines 304-306: . . .captured the signal of the sea-ice retreat that occurred shortly before. . . Sea ice retreat actually took place weeks and months before so it is not appropriate to say shortly before. The fact that IP25 was still detectable would be more likely the result of advection or resuspension.

Lines 378-380: However, it would then remain elusive why such an aged* land-derived influence was not visible in the river-dominated LS waters while it affected the sea-ice dominated region. Is it that elusive? It is puzzling that the authors did not consider that the presence of this land-derived material is likely the result of the release of material that was trapped in the ice during its formation on the shallow shelf. The trapped material is transported towards the outer shelves and released during ice melt, which was occurring at the time of sampling. This is an important and well-known process on the Siberian shelves. The interpretation must be improved to consider these ice-released particles. *How old? Be more specific.

Lines 394-396: Hence, results suggest a heterotrophic environment in the outer LS open waters where the river-derived DOC is transferred to relatively higher trophic levels via microbial incorporation (i.e, microbial loop). This sentence reflects a poor comprehension of the food web. Energy is not transferred to higher trophic levels through the microbial loop.

Table 1

What is TN? Mean sea ice percentage is over which area?

Table 2

This table does not belong in this manuscript.

Table 3

This qualitative analysis is nearly useless. The authors should definitely invest in quantitative taxonomic analyses to support their results.

Fig. 1

Switch North up.

Fig. 2

Should be removed, presented in other submitted paper.

Fig. 4

Patterns are often not as clear as described by the authors in the results/discussion. Be careful when interpreting.

Fig. 6

The new results should also be presented as whisker boxes for consistency. In the caption: East Siberian Sea, not Eastern Siberian Sea.

Fig. 7

Why only for East Siberian Sea?

---

## Author Comment (AC2) · 3 Aug 2017

We also apologize to the reviewer as some sentences looked odd due to the poor proofreading after that several people have worked on the same document with track change. These broken sentences were fixed and the manuscript has this time been proofread by a native English speaker.

**REV#1**

*Carbon geochemistry of plankton-dominated supra-1 micron samples in the Laptev and East Siberian shelves: contrasts in suspended particle composition.*

*General comments*

*This study want to improve the understanding of the chemical composition of plankton that dominates regions of the Arctic Ocean characterized by different sea-ice coverages and and in the ice-free Laptev Sea. The authors conclude that terrestrial carbon influence the POM in the Laptev Sea with higher influence of the River Lena. In the East Siberian Sea with ice-cover the influence of land is smaller. This is a valuable study in an important part of the ocean with a paucity of observations.*

*The methodology for 13C and 14C analyses seem adequate. However, this is not the case for the estimates of plankton diversity based on qualitative data from scanning electron microscopy. Either the authors have to convince us that this method is adequate and provide quantitative data analyzed by proper statistical tests. Otherwise markedly constrain conclusions regarding phytoplankton diversity or remove this entirely, perhaps using references to other studies instead.*

*Also the 13C and 14C analyses consistently lack objective statistical tests to support the conclusions made. Even if this data in general looks convincing this needs to be added.*

*The text is quite OK but there are some sentences which are not currently understandable. It does not appear that the last version has been checked by an English speaking person, which is recommended.*

*Provided that these remarks and important specific comments below are remedied I recommend that the manuscript is published after a major revision.*

*Specific comments*

*1) Title: Please revise the title replacing the "supra1-micron samples" term with "POM".*

We have replaced the term *supra micron* with "particulate organic matter (>10μm)" throughout the text

*2) r.28-36 I suggest to make the introductory paragraph shorter motivating the addressed question and the overall design of the study.*

We have shortened the abstract as requested.

*r.31-32 Unclear why "supra-" is used at this stage. Not a common term in my mind. Please just state "..the larger than 10 μm particulate organic matter (POM) fraction…" in the abstract. It´s clear from your definition what fraction that is covering.*

Changed throughout the text consistent with the title (see above)

*r. 37-41 These conclusions need to be better supported by statistical tests as specified in the result and discussion section.*

We ran a two-tailed T-test (homoscedasticity) and a Welch T-test (heteroskedasticity) to assess whether the differences between Laptev Sea and East Siberian Sea were statistically significant (see Methods). The text was changed accordingly when discussing differences between regions (e.g. line 324, 326, 353, 360).

*r. 42 "..communities via microorganisms". There are not reason to indicate several "loops". Please write concise and avoid unnecessary terms for clarity.*

*The term "loops" was removed according to the reviewer's suggestion*

*r. 44 The methodology does not seem adequate to assess the changes in diversity. Se comment in the result and discussion section. Also unclear what you mean by "…which is confirmed". Please rephrase.*

We agree with this comment. We toned down the part that deals with the characterization of algal assemblages. This is no longer part of the abstract and conclusions. However, we would like to leave Fig. 5 and some brief discussion in the text just to provide qualitative information.

*r. 45-46 Unclear what is meant by "…follows the general growth vs CO2aq supply model…". Please present in a clearer way.*

Rephrased to make it clearer (line 49-52).

*r-48-50. What basis is this prediction based on? Please add a specification.*

The destabilization of the Arctic cryosphere encompasses several aspects including sea-ice, glaciers and permafrost soil. Specific references were provided in the Introduction (line 63-84) which now contains even more details about the study region. Here in the abstract we can only report a summary of the future trajectories in response to the climate change.

*r. 65 What uptake are you referring to? Please rephrase and clarify.*

It refers to the CO2 uptake. Sentence rephrased (line 74).

*r. 67 "…also project to water-…" Does not seem like proper English. Not understandable. Please rephrase.*

The sentence was fixed (line 77).

*r. 92-93 if the term supra- is not earlier defined and internationally agreed upon I see no reason to use it here. Just use POM and define the size fraction studies for simplicity.*

We have replaced the term supra-micron fraction with POM throughout the text as suggested

*r. 96. "..characterized by bulk…" If the MS has not been language checked by English speaking persons or dedicated companies please do so.*

Again, this was one of those broken sentences resulting from poor text proof reading after several co-authors worked with track changes (line 107). All these odd sentences were corrected

*r. 92-101. This paragraph should be moved to the methods section.*

This is just a brief overview which we would like to keep in the Introduction. The Method section provides further and more specific details on the methods used.

*r. 106 Please start with presenting the studied Sea areas and their characteristics. Please consider to use a map with sampling sites.*

The map with cruise track and sampling site is already provided in the manuscript (Fig. 1). We added a section after the introduction to introduce the major features of the study region (section 2, line 114-129).

*r. 112 Pleas add how the Falcon tubes were cleaned.*

The method section was updated to include the cleaning procedure of the Falcon tubes (line 142-143).

*r. 119 How long time after sampling was the analysis? Was samples frozen all the time to analysis?*

The samples were kept frozen until lab analyses (line 141).

*r. 166 How was samples taken and preserved? Were they concentrated in some way? Please add. Is there any reason why not other autotrophs like flagellates, picoplankton and cyanobacteria were included?*

Samples were collected via large volume filtration of a nylon mesh. This was already well described in the text. We added further details about the samples collections (139-142). The 10um cut-off was chosen to avoid collecting fine terrigenous material in suspension.

*r. 170 What was the number of cells counted and precision of counting per sample?*

As we said this is a qualitative method which provides a snapshot of the dominant assemblages. It is not meant to be a statistical analysis. Thus, we agree with the previous comment and we have toned down this part.

*r. 184 Please add the accuracy and precision of the measurements of CO2 and δ13 CCO2.*

We added the precision for both CO2 and δ13C-CO2. However, this section was moved to the supplementary material.

*r. 229 Please present (for all variables) some confidence intervals or test, validating what are statistically significant differences between stations (i.e. accounting for spatial and short term temporal variability). E.g. if you want to claim differences between sea areas show by a proper statistical methods that they are different from each other.*

We used the T-test between ice-free and ice-dominated regions while Pearson correlation was used to investigate the relationship among variables (see Methods section 3.5). However, we mainly focused on the new data presented in this study (dual-carbon isotope and biomarker data). The rest of the data regarding the surface water properties were elaborated, discussed and interpreted in other studies (Humborg et al., 2017; Salvado et al., 2016 ).

*r. 236 The data referred to in Humborg et al. need either to be published before accepting this paper, or data presented in this paper.*

Humborg et al has been just accepted in *Global Biogeochemical Cycles DOI: 10.1002/2017GB005656*

*r. 243 How do you define depletion? Please be more specific and refer to comparative data or references. Similar for "low" at r. 245.*

This sentence was modified according to this comment (line 286-289).

*r. 255 Please specify what "margin" that is referred to. This sentence is not possible to understand. Please rephrase.*

We have replaced "margin" with "continental margin" for clarity (line 300).

*r. 259 It´s not obvious how the concentration of lignin or hydroxyl fatty acids will say anything about effects on the POM fraction. Do you mean the conc. of these compounds in the POM? What about many other effects on living POM like species composition and functionality like growth or edibility? Do assume that most POM is non-living? Please motivate you analyses better relative the aim.*

As stated in the text, lignin phenols and cutin acids are uniquely produced by terrestrial vegetation. Therefore, these analyses were carried out to assess whether or not the samples were affected by land-derived material (i) directly supplied by rivers, (ii) trapped in the sea-ice and (iii) resuspended from the sediment. The concentrations of lignin phenols are close to the detection limit while cutin were not detectable. This implies that the material collected is primarily autochthonous marine POM.

*Fig.3 Please add what error bars are showing. Please specify what the values are relative against (carbon or mass?).*

This figure shows the CuO oxidation fingerprint of the samples. Upon Cuo oxidation organic matter releases different biomarkers whose composition provide information about the source. The percentage refers to the total CuO oxidation yields and the error bar reflects that natural variability observed in different carbon pools. For example, soils are rich in lignin phenols and cutin acids while phytoplankton batch cultures upon CuO oxidation don't yield these terrestrial biomarkers. By contrast, they produce a large amount of low molecular weight fatty acids consistent with the POM collected in this study. We added further details in the caption of Fig.3.

*Samples*

*r. 276 Support you statement with a tests showing that these are different. What do a base the "high" and "low" assessments on (relative what)?*

As previously mentioned, we used a T-test/Welch T-test to show that these differences are statistically significant ($p < 0.01$) (line 324)

*r. 275-287 SEM is not a proper method to asses diversity of phytoplankton (or present quantitative SEM data from sufficient number of samples?). That the diversity of phytoplankton is different between sea areas is therefore not sufficiently well demonstrated. Concentrations of at least major taxonomic groups is requested based on microscopy counting with adequate methodology (e.g. sediment chambers and reverse light microscopy) and statistical precision presented. Preferably also including flagellates, picoeucaryotes and cyanobacteria. Established diversity index should be used and tested for difference.*

As previously mentioned, we have used this method to provide a general overview of the dominant taxa. We agree with the reviewer on this comment and we are aware that this procedure has only qualitative applications. We would still like to keep this part but make clear in the text that this is only qualitative information. Any reference in the abstract and conclusions about the phytoplankton taxa were removed from the revised text.

*r- 288-289 Unclear how a line can detect a bloom. Please be more specific.*

The sentence was corrected (line 336).

*r.293-294 I don´t agree that the presented data convincingly show that dinoflagellates were dominating. How is the SEM preparation influencing different phytoplankton species? Is there a selections for robust dinoflagellate shells? Provide a reference or control experiment clarifying this.*

There is no reason to think of selective preservation for different taxa during sampling and microscope analysis as all the samples have been treated in the same way. Despite the fact that a taxa quantification would provide much better statistical evidence, these results would be consistent with the qualitative investigation done here, in particular the large difference between LS (rare diatoms) and ESS (diatom dominated). Again, we took several SEM images for each sample which are absolutely consistent with what observed with the optical microscope.

*r. 297-301 As presented isn´t IP25 then specifically indicating presence of sea-ice diatoms, not "..sampling of different plankton taxa…? Also can other sources of IP25 have contributed to the variability. Consider a re-interpretation. I suggest to calculate if the found conc. of IP25 could come from an expected conc. of diatoms in the sea ice and present that.*

IP25 detects specific sea ice diatoms (Pleurosigma stuxbergii var. rhomboide, Haslea crucigeroides (and/or Haslea spicula) and Haslea kjellmanii, Brown et al., 2014b) which account for only a minor fraction of the sea-ice taxa. No other sources (e.g. from land) supply IP25 expect from these sea-ice taxa. To the best of our knowledge, the actual end-member doesn't exist and it likely varies depending on the concentration of these aforementioned species in the sea-ice

*r. 309-310 Please provide a statistical test showing a significant trend of CO2 and δ13 CCO2.*

*p* values were added in both linear correlations (Fig.7).

*r. 316-317. On what basis is it assumed that the present dinoflagellates are hetero- or mixotrophic? That some dinoflagellates can eat bacteria is well shown in the literature. However, not that they are significant consumers of diatoms? Please provide a reference for this if so.*

Our hypothesis is based on the dual-carbon isotope fingerprint. Specifically, out of the different carbon source known in the region (Fig.6), the dual-carbon isotope signature of the POM is consistent only with the Lena river DOC (and rather different from the Lena river POM). Thus, in our hypothesis, dinoflagellates feed on bacteria that develop in the terrestrial DOC-rich plume of the Lena river.

*r. 324-325 I suggest to rephrase to "…, supporting the importance of terrestrial DOC as a carbon source for the food web in the river plume…. ).*

Text changed according to this suggestion (line 399-400).

---

## Author Comment (AC3) · 3 Aug 2017

*General comments*

*The objective of this study was to investigate the composition of the suspended particulate organic matter in ice-covered and ice-free waters over the Laptev and East Siberian shelves. The main problem of this study is to assume that these samples are plankton-dominated, as indicated by the title. There are no data to support the fact that phytoplankton dominated the suspended particulate matter and such a dominance would actually be quite surprising over the shallow Siberian shelves (a lot of particulate material is resuspended and/or transported with the ice). An effort should be made to quantify the phytoplankton contribution and composition before resubmitting this manuscript. If the authors somehow collected ice samples during this expedition, it would be relevant to compare the composition of the particulate matter in the ice with the composition of the suspended matter.*

*Another important problem is that too much of the current manuscript is based on another paper submitted elsewhere by many of the same authors that seems to be very similar to the current manuscript. This problem must be addressed. Overall, while the study had the potential to provide interesting results from a very rarely sampled region, the current results do not bring very interesting or new information. It is well-known that ice covered regions are productive and display high concentrations of particulate matter. The interpretation of the results must be reevaluated in this context. Also, please keep in mind and specify throughout the manuscript that these are late summer observations and that conditions may be quite different during the productive spring period. Finally, the manuscript is too long, often repetitive, and the text needs to be revised by a native English speaker.*

To the best of our acknowledge, this is the first study that have characterized the dual-carbon isotope fingerprint of plankton-rich samples in this region. Knowing the isotopic composition of plankton is relevant for several applications which were listed in the text. For instance, source apportionment models in the study area have historically relayed on data collected in other Arctic regions. This is why we think the results obtained are worth and important for the community working on the carbon cycling in this area. The timing of the phytoplankton blooming is well discussed in the text and results were interpreted accordingly

*Specific comments*

*Title*

*I have never heard the terms supra-micron or supra-POM and I don't think there is a need for it. Please remove the term supra- throughout the manuscript.*

As mentioned for the other reviewer, we have replaced the *"supra-micron POM"* term with POM (>10µm)

*Abstract*

*Lines 50-51: Comments like these are not informative. Always be specific.*

We followed the suggestion. The paragraph ends with specific details about changes in the study region (line 53-57)

*Introduction*

*Lines 54-55: Provide more recent references.*

We have updated the references with recent studies dealing with the sea-ice retreat in the Arctic Ocean (line 64)

*Material and methods*

*Some information on the dates of sampling are required.*

*Lines 110-113: Several steps are unclear. When it is mentioned that particulate material was kept frozen, it means the filters? In which state were the samples transferred into the centrifuge? Were the samples thawed first?*

This part was edited to make it clearer (line 137-141)

*Lines 115-116: Such information belongs in figure captions.*

Removed

*Lines 150 and 154: IP25 is a highly branched isoprenoid mono-unsaturated alkene. Introduce it properly and only once.*

Text changed accordingly (line 180-184).

*Section 2.3 Microscopic images of plankton\* This is probably the biggest shortcoming of the study. It is baffling that the authors use microscopic images as a qualitative tool but did not include a quantification of the different phytoplankton groups. This would definitely improve the quality of the study. \*Always precise if it is phytoplankton or zooplankton. Plankton is not a term precise enough.*

Of course, we agree that quantitative information would have been more informative. Ours is essentially a biogeochemical study and these snapshots obtained via SEM and transmitted light microscope, provide only complementary information. For this study, we took several SEM images (in the paper we show just one magnified example) about 10 per station randomly sampled in the freeze-dried material. This, combined with traditional microscope analyses, allowed us to see major trends within the dataset, for example we could hardly see any diatoms in the Laptev Sea while further east diatoms were dominant. Thus, despite the limitation of our approach (which we acknowledge) we still believe that this is relevant information. We decided to tone down this part in acknowledgement to the reviewer's comment but still keep it in the discussion. In the revised manuscript, we removed any comment on phytoplankton taxa in both abstract and conclusion. However, we would still like to keep this part in the discussion making sure that the reader understands the qualitative applications of our approach

*The material and methods section is too long and often repetitive. Reduce.*

We went through this section but couldn't find any redundant information. However, methods on CO2 measurements were moved to the supplementary materials

*Results/discussion*

*Section 3.1. Surface water conditions Most of this section does not belong in the paper. All the results (salinity, temperature, nutrients...) for which material and methods were not presented in the precedent section must be removed from the manuscript and the figures/tables as well. This is even more crucial considering that the same results are part of another submitted paper from the same authors. It is not appropriate to submit the same results twice and all the results that were submitted in Humborg et al. must be removed. Instead the authors should refer to these results in the discussion, which would be much stronger if or once the other paper is published. It would be more appropriate to start the discussion with section 3.2.*

Humborg et al has been accepted in *Global Biogeochemical Cycles (and updated in our ref list)*. Following the reviewer's suggestion, the discussion now starts with section 4.1 and the former 3.1 section dealing with the surface water properties has become section 3 ("*Surface water conditions during the SWERUS-C3 expedition*)

*Lines 220-222: You should never write sentences in this form: Figure 1 displays...*

This sentence was changed according to the reviewer comment (line 262)

*"Table 1 reports..." This is the type of mistakes made at the undergraduate level.*

Specifically for the above comment, please refer to point 3 of the "General obligations for referees" document on the *Ocean Science* website. Thanks

*Line 225 and others: All maps (figures 1, 2 and 4) should be switched with North towards the top to help with the description of the results. This is the usual and correct way to place a map and it is less confusing when looking for the westernmost stations.*

We must disagree on this point. In this special issue, there are at least 8 manuscripts (Miller et al., Anderson et al., o'Regan et al, Björn et al, etc) with the exact same polar projection. In fact, this is the default ESRI Arcgis projection consistent with the IBCAO format (Jakobsson et al., 2012.GRL). We will keep this format

*Lines 230-232: DOC concentrations mirrored... 'Mirrored' does not mean the opposite, it means similar. Get an English speaker to review your paper. And please limit your use of the word 'thus'.*

Of course, we did not mean similar in terms of absolute values. We were just commenting on the general spatial distribution as DOC is clearly affected by the river plume: low salinities are associated with high DOC concentrations. Sentence reformulated (line 273)

*Line 252: It is late and unnecessary to introduce the term TerrOC at this point. Either you introduce it earlier or you use other terms for consistency.*

TerrOC was removed and replaced with "terrestrial organic carbon"

*Lines 284-287: These phytoplankton species are typically observed late in the season. This should be specified. Chaetoceros and Thalassiosira are pelagic species growing in water only while Fragilariopsis cylindrus and oceanica grow both in ice and water (they are not sea ice species necessarily). More information could be obtained through extensive and quantitative taxonomic analyses of the existing samples.*

Again, we agree with the reviewer that this minor part of our ms has severe limitations but we believe it still provides some useful general information on the dominant trends within the study region (see previous comments). For example, we can clearly see a marked difference between the Laptev Sea (no diatoms with abundant dinoflagellates) and the East Siberian Sea (dominated by

diatoms) regardless of the method used (SEM vs traditional microscope). Again, our analysis is based on several SEM images coupled with optical microscope slides

*Lines 304-306: … captured the signal of the sea-ice retreat that occurred shortly before… Sea ice retreat actually took place weeks and months before so it is not appropriate to say shortly before. The fact that IP25 was still detectable would be more likely the result of advection or resuspension.*

Corrected according to this suggestion. Resuspension would bring lignin and other wax lipids (cutins) which, however, were not detected in the POM samples. Advection from surface waters might be a reasonable hypothesis though. Text changed accordingly (line 355-356)

*Lines 378-380: However, it would then remain elusive why such an aged\* land-derived influence was not visible in the river-dominated LS waters while it affected the sea-ice dominated region. Is it that elusive? It is puzzling that the authors did not consider that the presence of this land-derived material is likely the result of the release of material that was trapped in the ice during its formation on the shallow shelf. The trapped material is transported towards the outer shelves and released during ice melt, which was occurring at the time of sampling. This is an important and well-known process on the Siberian shelves. The interpretation must be improved to consider these ice- released particles. \*How old? Be more specific.*

It's not puzzling at all as we specifically used organic biomarkers to trace the land-derived material. Figure 2 shows what type of CuO oxidation fingerprint you would get in a case of land-derived influence. For example, lignin phenols are clearly dominant in terrestrial soil samples while they are close to detection limit in our POM samples. Cutin derived products (waxes on plant leaves) were not even detected in the POM samples. As stated in the text, lignin phenols and cutins are exclusively produced on land and their negligible concentration in POM samples implies insignificant terrestrial influence. In other words, the samples are dominated by marine material. In fact, POM samples are consistent with the CuO oxidation fingerprint of algal batch cultures which mainly yield fatty acids, dicarboxylic acids, p-hydroxy phenols and benzoic acids (in this order of abundance) (Goni and Hedges, 1995)

Through this comment, we have realized that we did not mention the particulate transport by fast ice in the text. To provide a better picture regarding the sediment transport in the study region, we added a sentence about this mechanism (line 303)

*Lines 394-396: Hence, results suggest a heterotrophic environment in the outer LS open waters where the river-derived DOC is transferred to relatively higher trophic levels via microbial incorporation (i.e, microbial loop). This sentence reflects a poor comprehension of the food web. Energy is not transferred to higher trophic levels through the microbial loop.*

Here we refer to the terrestrial carbon being transferred from the DOC to the heterotrophic community via bacteria present in the Lena river plume. According to the radiocarbon signature (Fig. 6) of the Laptev samples, we infer that the DOC from the Lena is taken up by the microbial

communities on which other heterotrophic communities (e.g. *Protoperidinium* spp;) feed on. Despite the fact that the samples have been collected in a region affected by the Lena plume (see salinity and DOC data, Fig 2), our modern 14C signature of POM largely differs from the particulate material supplied by the Lena river characterized by a 14C depleted signature (Fig. 6). In contrast, the POM signature seems to be more consistent with the Lena DOC fingerprint (Fig. 6). This would also explain the depleted stable carbon isotope composition despite the negligible terrestrial influence (biomarker results). It's also worth mentioning that DOC in the outer Laptev Sea is over one/two order of magnitude higher than the POC (Humborg et al., 2017; Salvado et al 2016)

*Table 1 What is TN? Mean sea ice percentage is over which area?*

TN is total nitrogen. Data are not discussed so they were removed in the new version

*Table 2 This table does not belong in this manuscript.*

As previously mentioned, these data are presented only with the intention of contextualizing our results. The discussion has been rearranged following the reviewer's suggestion. We now start the discussion from the new organic geochemistry data. Showing complementary data is a common procedure when studies are part of a multidisciplinary expedition during which research teams measured different parameters. Humborg et al has been recently accepted and properly cited in the manuscript

*Table 3 This qualitative analysis is nearly useless. The authors should definitely invest in quantitative taxonomic analyses to support their results.*

See comments above

*Fig. 1 Switch North up.*

See comments above

*Fig. 2 Should be removed, presented in other submitted paper.*

See comments above

*Fig. 4 Patterns are often not as clear as described by the authors in the results/discussion.*

*Be careful when interpreting.*

As just mentioned above, we have divided the study area into two sub-regions (Laptev Sea_open waters and East Siberian Sea-ice-dominated) and carried out a T-test as suggested by reviewer#1 to show whether or not the differences are statistically significant

*Fig. 6 The new results should also be presented as whisker boxes for consistency. In the caption: East Siberian Sea, not Eastern Siberian Sea.*

Caption corrected. The whisker plots presented here are used to summarize large dataset. For example the ICD end-member is made of 301 radiocarbon data. However, we don't think we can do the same for the POM samples in each considering the limited number of radiocarbon values

*Fig. 7 Why only for East Siberian Sea?*

Because only the East Siberian Sea is an autotrophic system where CO2 is actually consumed by biological activity (i.e., depletion compared to the atmospheric value) (Humborg et al., 2017)

---

## Author Response (AR2)

We thank the Topic Editor Leif Anderson for his time and comments on the revised text os-2017-21. We have fixed all the minor points raised in his review. We have rewritten a couple of sentences in the abstract, replaced Fig.2 as suggested and fixed all the typos throughout the text including the list of references as well.

We think that the paper has greatly improved and is now ready for publication in Ocean Science.